# Genetic variation in adaptability and pleiotropy in budding yeast

Elizabeth R Jerison[1,2,3†], Sergey Kryazhimskiy[4†], James Kameron Mitchell[2], Joshua S Bloom[5], Leonid Kruglyak[5], Michael M Desai[1,2,3]*

[1]Department of Organismic and Evolutionary Biology, Harvard University, Cambridge, United States; [2]Department of Physics, Harvard University, Cambridge, United States; [3]FAS Center for Systems Biology, Harvard University, Cambridge, United States; [4]Section of Ecology, Behavior and Evolution, Division of Biological Sciences, University of California, San Diego, San Diego, United States; [5]Department of Human Genetics, University of California, Los Angeles, Los Angeles, United States

**Abstract** Evolution can favor organisms that are more adaptable, provided that genetic variation in adaptability exists. Here, we quantify this variation among 230 offspring of a cross between diverged yeast strains. We measure the adaptability of each offspring genotype, defined as its average rate of adaptation in a specific environmental condition, and analyze the heritability, predictability, and genetic basis of this trait. We find that initial genotype strongly affects adaptability and can alter the genetic basis of future evolution. Initial genotype also affects the pleiotropic consequences of adaptation for fitness in a different environment. This genetic variation in adaptability and pleiotropy is largely determined by initial fitness, according to a rule of declining adaptability with increasing initial fitness, but several individual QTLs also have a significant idiosyncratic role. Our results demonstrate that both adaptability and pleiotropy are complex traits, with extensive heritable differences arising from naturally occurring variation.

DOI: https://doi.org/10.7554/eLife.27167.001

*For correspondence:
mdesai@oeb.harvard.edu

†These authors contributed equally to this work

## Introduction

All organisms can evolve in the face of environmental perturbations. Yet even closely-related individuals may differ in how quickly and effectively they adapt (*Kirschner and Gerhart, 1998*; *Masel and Trotter, 2010*; *Lauring et al., 2013*). Genetic modifiers that create these differences in evolvability are subject to natural selection, and the potential consequences of secondary selection on evolvability has been the subject of extensive theoretical work (*van Nimwegen et al., 1999*; *Wilke et al., 2001*; *Masel and Bergman, 2003*; *Ciliberti et al., 2007*; *Cowperthwaite et al., 2008*; *Draghi et al., 2010*; *Soyer and Pfeiffer, 2010*). Only recently, however, have experiments begun to identify these genetic modifiers and characterize their effects (*Montville et al., 2005*; *Jarosz and Lindquist, 2010*; *Woods et al., 2011*; *McDonald et al., 2012*; *Stern et al., 2014*; *McDonald et al., 2016*; *Xiao et al., 2016*).

Modifiers of evolvability act by changing parameters of the evolutionary process. For example, mutator or antimutator alleles often spread through adapting laboratory populations (*Sniegowski et al., 1997*; *McDonald et al., 2012*; *Wielgoss et al., 2013*). Other empirical work has shown that alleles that modify recombination rates can speed the rate of adaptation or help purge deleterious load (*Zeyl and Bell, 1997*; *Colegrave, 2002*; *Goddard et al., 2005*; *Cooper, 2007*; *Becks and Agrawal, 2012*; *Gray and Goddard, 2012*; *McDonald et al., 2016*; *Xiao et al., 2016*). Mutations can also affect evolvability in more subtle ways, through epistatic interactions that shape the future directions that evolution can take. These epistatic effects create historical contingency:

the presence of an initial mutation changes the fitness effects of other potential mutations, opening up or closing off future adaptive trajectories (*Blount et al., 2008*; *Woods et al., 2011*; *Salverda et al., 2011*; *Blount et al., 2012*).

The identification and characterization of epistasis among specific sets of mutations has been a subject of intensive recent research. For example, several groups have analyzed combinations of mutations within the same protein that fixed during evolution (*Weinreich et al., 2006*; *Bridgham et al., 2009*; *Gong et al., 2013*; *Weinreich et al., 2013*; *Palmer et al., 2015*), or combinations of multiple mutations in a region of a protein (*Hietpas et al., 2011*; *Araya et al., 2012*; *Jacquier et al., 2013*; *Harms and Thornton, 2014*). These studies often find interesting intra-protein patterns of epistasis, including sign epistasis (a mutation shifts from beneficial to deleterious or vice versa depending on the background genotype) or higher-order epistasis (the combined effect of a set of mutations is not fully explained by the effects of any subset of these mutations). Other studies have measured pairwise epistasis among large collections of genome-wide mutations (e.g. all single gene knockouts) (*Jasnos and Korona, 2007*; *van Opijnen et al., 2009*; *Costanzo et al., 2010*; *Szappanos et al., 2011*; *Babu et al., 2014*; *Costanzo et al., 2016*). Still others have analyzed epistasis among all combinations of a few mutations that accumulated along the line of descent in a microbial evolution experiment (*Chou et al., 2011*; *Khan et al., 2011*; *Wünsche et al., 2017*), measured the fitness effects of individual mutations in divergent genetic backgrounds (*Wang et al., 2016*), or identified examples where a single mutation opens up an opportunity to colonize a new metabolic niche (*Blount et al., 2008*; *Blount et al., 2012*; *Quandt et al., 2014*; *Quandt et al., 2015*).

This body of work has extensively analyzed genetic interactions between individual mutations, a phenomenon we call *microscopic epistasis* (*Good and Desai, 2015*). These interactions can lead to historical contingency in the order and identities of mutations that spread in an adapting population (*Weinreich et al., 2005*). However, to affect evolvability a mutation must change the entire distribution of fitness effects of future mutations. That is, it must lead to a region of the fitness landscape which has statistically different properties, a phenomenon we refer to as *macroscopic epistasis* (*Good and Desai, 2015*).

Microscopic and macroscopic epistasis are distinct but not mutually exclusive properties of the fitness landscape. For example, a particular pattern of microscopic epistasis between individual mutations might (or might not) change the overall shape of the distribution of fitness effects, and therefore may (or may not) lead to macroscopic epistasis. Conversely, macroscopic epistasis can reflect microscopic epistasis among specific loci, but could also arise in the absence of microscopic epistasis if the pool of adaptive mutations is small and acquiring any individual mutation depletes future opportunities for further adaptation.

Although it is clear that microscopic epistasis is widespread, much less is known about patterns of macroscopic epistasis. Thus it is unclear how often and which mutations lead to differences in evolvability. Several recent studies have addressed this question by analyzing variation in the rate of adaptation among closely related microbial strains (*Couce and Tenaillon, 2015*). These studies have found that a few initial mutations can affect how quickly a strain adapts in the future (*Barrick et al., 2010*; *Perfeito et al., 2014*; *Kryazhimskiy et al., 2014*; *Szamecz et al., 2014*). Much of this variation in adaptability can be explained by the fitness of the founder, with fitter founders adapting more slowly (i.e. the rule of declining adaptability).

This work suggests that the rate at which a strain adapts is a heritable trait, influenced not just by stochastic evolutionary factors, such as the random occurrence and fixation of mutations, but also by genotype. However, little is known about variation in evolvability across larger genetic distances. In other words, how different is the evolutionary process starting from very different points in genotype space? Does a systematic pattern of declining adaptability with increasing fitness still apply among more distantly related strains? Or are differences in evolvability caused primarily by specific interactions between any of the initial genetic differences and the pool of potential future mutations? Additionally, previous studies have analyzed evolution in a single environmental condition, and thus could not test how initial genotype might affect the pleiotropic consequences of evolution (i.e. how adaptation in one environment affects fitness in other conditions). Note that here we use the term 'pleiotropy' in a sense that is common in experimental evolution but not in the genetics literature: it reflects how mutations that affect one trait (fitness in one environment) also influence other traits (fitness in different environments).

In this study, we address these gaps by measuring genetic variation in adaptability among 230 haploid *S. cerevisiae* genotypes ('founders') derived from a cross between two substantially diverged parental strains (*Bloom et al., 2013*). The parental strains (a laboratory strain, BY, and a wine strain, RM) differ at ~50,000 loci. Thus our 230 founders differ by on average ~25,000 mutations that are to some degree representative of natural variation in yeast. We evolve four replicate lines descended from each founder for 500 generations in each of two different laboratory environments. As in earlier studies, we can then measure variation in evolutionary outcomes that can be attributed to founder identity, above and beyond inherent evolutionary stochasticity. However, because our founders are chosen from a panel of strains designed for mapping the genetic basis of phenotypes of interest — each founder contains a random subset of the genetic variation between RM and BY strains — we can also use standard methods from quantitative genetics to identify specific loci that affect adaptation.

We focus on quantifying two specific aspects of evolvability: the rate of adaptation of each founder in each environment, and the pleiotropic consequences of adaptation in one environment for fitness in the other. We also address the influence of founder genotype on the genetic basis of future adaptation. Our results demonstrate that variation in both adaptability and pleiotropy is widespread. Despite the fact that our founder genotypes span evolutionary distances that are several orders of magnitude larger than tested in earlier work, the variation in their rates of adaptation still follows an overall rule of declining adaptability. The variation in pleiotropic consequences of adaptation between founders is also partly governed by founder fitness. Above and beyond these effects of founder fitness, we also identified several specific quantitative trait loci (QTLs) that affect adaptability and pleiotropy, including one which has a large influence on the genetic basis of future evolution. Thus both macroscopic and microscopic epistasis are important in our system, and lead to systematic variation in the rate, genetic basis, and pleiotropic consequences of adaptation.

## Results

To measure genetic variation in adaptability in budding yeast, we selected 230 segregants ('founders') from a cross between a wine strain, RM, and a laboratory strain, BY (Materials and methods). The parental strains RM and BY differ by about 0.5% at the sequence level, so each founder differs from its siblings at about 25,000 loci across the genome. We chose to analyze evolvability in two environments: rich media at an optimal temperature of 30°C (the OT environment), and synthetic media at a high temperature of 37°C (the HT environment). We measured the initial fitness of each founder in each environment and found that the genetic differences between founders lead to substantial variation in starting fitness in both environments (*Figure 1*). This variation is strongly correlated, implying that at least on average, alleles that are more beneficial in our OT environment also tend to be more beneficial in our HT environment.

We established eight replicate lines from each founder (a total of 1840 populations), and evolved half of these replicate populations in our OT environment and half in our HT environment. After 500 generations of evolution, we measured the fitness of each descendant population in both environments (*Figure 1*). The difference between the initial fitness of the founder, $X$, and the fitness of a descendant population, $X'$, is a measure of the rate at which that population adapted.

The measured rate of adaptation includes contributions from experimental error (inaccuracies in our fitness measurements lead to apparent differences in the rate of adaptation), from evolutionary stochasticity (some populations are more lucky than others in accumulating beneficial mutations), and from the inherent adaptability of the founder (some founders may tend to adapt faster than others). We are primarily interested in understanding the third component: the effect of founder genotype on adaptability. To disentangle this from evolutionary stochasticity, we can compare the variation in the rate of adaptation between lines descended from the same founder with the variation between lines descended from different founders. Similarly, to disentangle the effects of measurement error, we can compare the variation between lines descended from the same founder with the variation between replicate fitness measurements of the same populations. We could in principle compare these different sources of variation using an analysis of variance (ANOVA) framework, following the approach we used in previous work (*Kryazhimskiy et al., 2014*). Here, we instead describe this analysis in the language of quantitative genetics. While some of the underlying

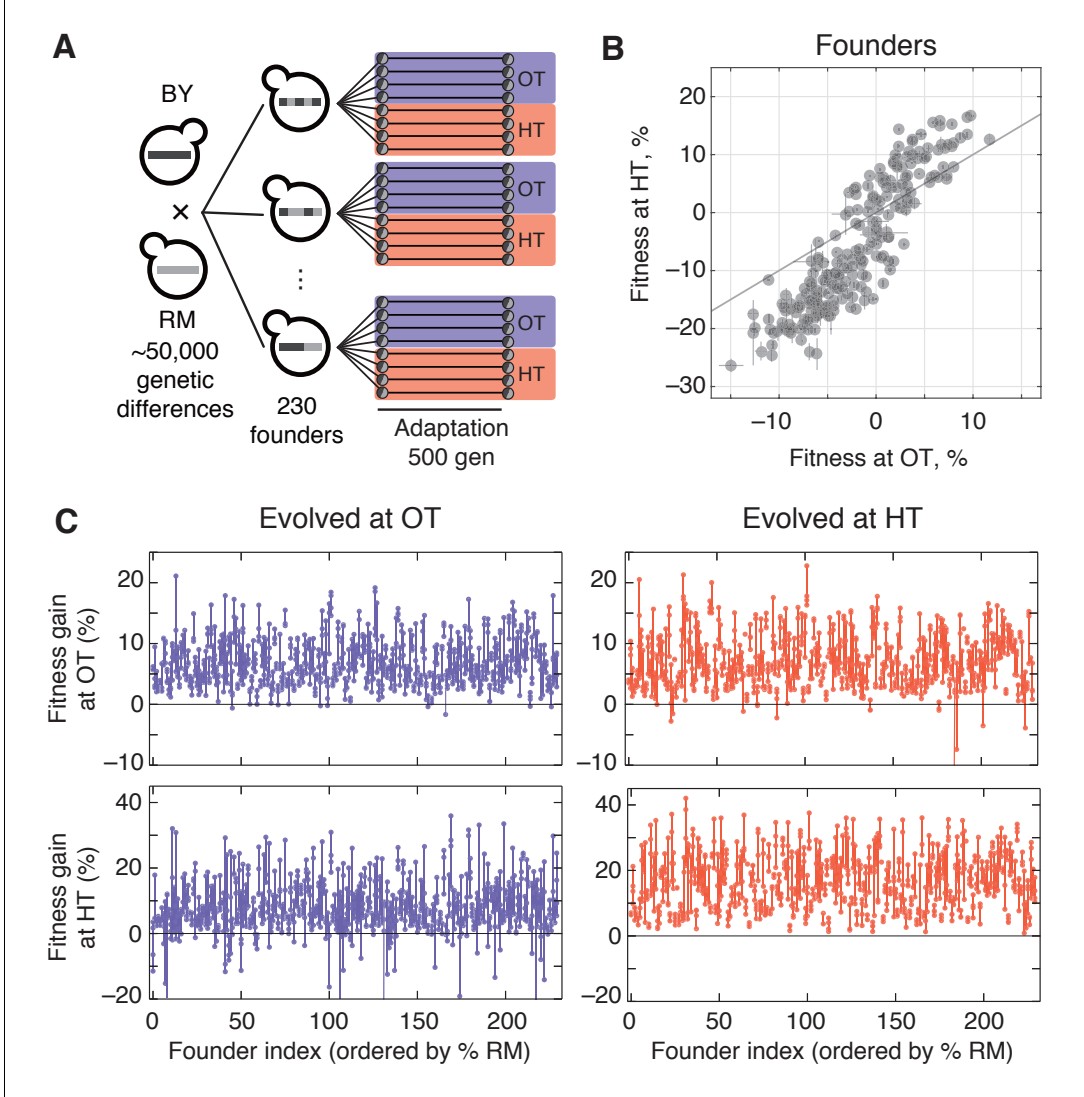

**Figure 1.** Experimental design and fitness gains in evolved populations. (A) Schematic of the experimental design. (B) Fitness of all 230 founders at both OT and HT. (C) Fitness of all evolved lines in both OT and HT environments, after 500 generations of adaptation in one of the environments. Each point represents the fitness of one population; vertical lines connect populations descended from the same founder. Founders are ordered by the fraction of their genome derived from the RM parent.

DOI: https://doi.org/10.7554/eLife.27167.002

methods are equivalent, this quantitative genetics framework provides a useful perspective, and makes it possible to address additional questions about the genetic basis of variation in adaptability.

We begin by calculating the broad and narrow sense heritability of founder fitness and adaptability. Here the broad-sense heritability of a phenotype, $H^2$, is the fraction of the variance in that phenotype that can be explained by genotype. The narrow-sense heritability, $h^2$, is the fraction of the total variance in the phenotype which can be explained by an additive model of the individual loci. We calculate both broad and narrow-sense heritability using standard methods (*Yang et al., 2010*; *Zuk et al., 2012*; *Visscher et al., 2006*), which exploit the fact that in an additive model the covariance between founder phenotypes is a linear function of their relatedness (Materials and methods). Since our populations are all haploid, the gap between broad-sense and narrow-sense heritability arises primarily from epistatic interactions between loci and any experimental error in our estimation of these quantities, although it may also carry contributions from untagged rare variants, mitochondrial copy number variation, and any uncontrolled $G \times E$ effects (*Bloom et al., 2015*).

We first consider the heritability of the founder fitness. We find that this has a broad-sense heritability near 100% (*Figure 2*, *Table 1*), indicating that measurement errors are small compared to genuine differences in fitness between founders. Consistent with previous work (*Bloom et al., 2013*), we find a substantial gap between $h^2$ and $H^2$ (*Figure 2*, *Table 1*), indicating the importance of epistasis in determining founder fitness.

We next consider the heritability of adaptability. The broad-sense heritability of adaptability, $H^2_{\Delta X}$, is the fraction of variance in $\Delta X = X' - X$ that can be attributed to founder genotype, rather than to inherent evolutionary stochasticity or to measurement errors. In contrast to founder fitness, which we expect to be a property of the genotype, there is no reason that the rate of adaptation must be heritable. However, multiple previous studies have found that genotype does affect the rate of adaptation in microbial populations (*Burch and Chao, 2000*; *Barrick et al., 2010*; *Woods et al., 2011*; *Perfeito et al., 2014*; *Kryazhimskiy et al., 2014*). Consistent with this earlier work, we find that adaptability is heritable, with $H^2_{\Delta X} \approx 0.61$ in the OT environment and 0.65 in the HT environment (*Figure 2*, *Table 1*). The fact that the values of $H^2_{\Delta X}$ are lower than those of $H^2_X$ is expected because, in addition to measurement errors, our measure of adaptability is affected by the inherent randomness of the evolutionary process.

As before, we fit a linear model to estimate the narrow-sense heritability of adaptability $h^2_{\Delta X}$. We find $h^2_{\Delta X} = 0.46$ in the OT environment and $h^2_{\Delta X} = 0.44$ at HT (*Figure 2*, *Table 1*). The difference between $H^2_{\Delta X}$ and $h^2_{\Delta X}$ indicates that about two thirds of the heritable variation in adaptability can be explained by an additive model of the underlying genotype, showing that epistasis is important to this trait (*Figure 2*).

## Predicting adaptability from genotype and founder fitness

We next sought to map the genetic basis of the observed variation in founder fitness and adaptability. To do so, we applied a standard iterative procedure to identify QTLs that affect each of these traits in each environment (see Materials and methods). We identified 6 QTLs that affect founder fitness at OT, and 3 QTLs that affect founder fitness at HT. Consistent with the strong correlation between these two phenotypes, 2/3 of the HT QTLs also appear as OT QTLs (*Supplementary file*

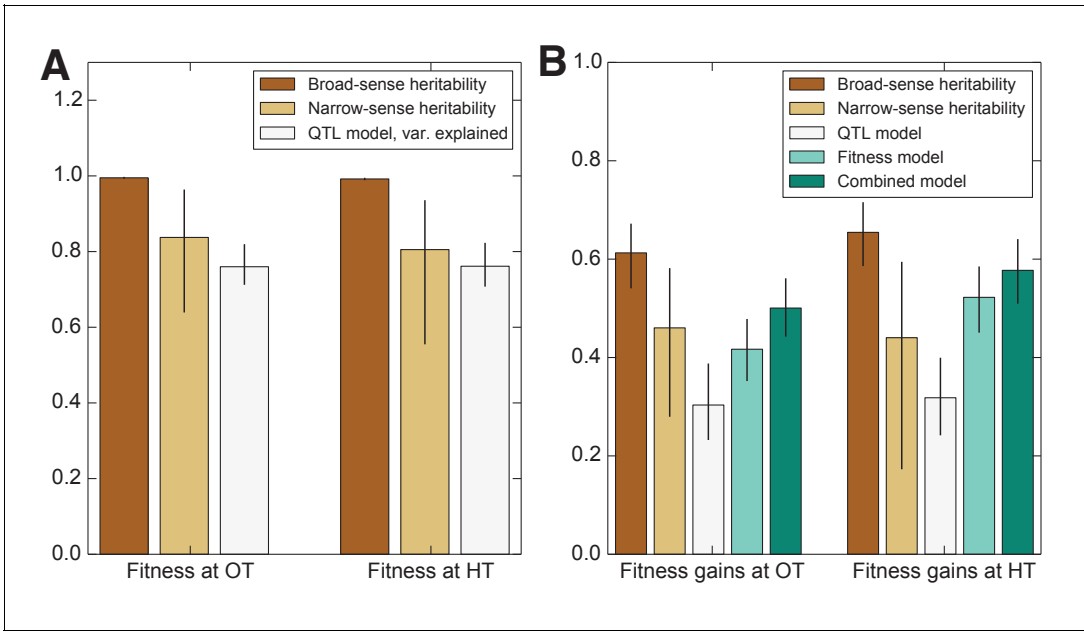

**Figure 2.** Heritability of fitness and adaptability. Tan bars denote broad-sense and narrow sense heritability of (**A**) founder fitness and (**B**) fitness gain after evolution in the indicated environment (i.e. adaptability). White bars denote the heritability explained by a linear model based on the QTLs identified as affecting initial fitness or adaptability (Materials and methods). Green bars indicate the heritability of adaptability explained by initial fitness in that condition. Error bars indicate 95% confidence intervals.
DOI: https://doi.org/10.7554/eLife.27167.003

**Table 1.** Heritability of fitness, adaptability, and pleiotropy, along with the fraction of variance explained by QTL, fitness, and combined models.

Row 5 represents the median added variance explained by the fitness model over the QTL model under a jacknife over segregants (Materials and methods); Row 7 similarly represents the added variance explained by the combined model over the fitness model. Paired numbers in parentheses denote 95% confidence intervals; single numbers in parentheses denote lower bounds (5$^{th}$ percentile) on added variances.

| | Fitness at OT | Fitness at HT | Adaptability at OT | Adaptability at HT | Pleiotropy HT pops at OT | Pleiotropy OT pops at HT |
|---|---|---|---|---|---|---|
| $H^2$ | 0.995 (0.992, 0.997) | 0.992 (0.988, 0.994) | 0.613 (0.541, 0.672) | 0.654 (0.586, 0.716) | 0.622 (0.552, 0.680) | 0.291 (0.193, 0.394) |
| $h^2$ | 0.837 (0.639, 0.964) | 0.805 (0.555, 0.936) | 0.460 (0.280, 0.582) | 0.440 (0.173, 0.595) | 0.477 (0.308, 0.636) | 0.091 (0, 0.228) |
| $r^2$, QTL models | 0.760 (0.712, 0.820) | 0.761 (0.707, 0.823) | 0.303 (0.232, 0.388) | 0.318 (0.242, 0.340) | NA | NA |
| $r^2$, founder fitness | NA | NA | 0.417 (0.352, 0.478) | 0.522 (0.451, 0.585) | 0.425 (0.361, 0.492) | 0.127 (0.082, 0.182) |
| Added variance fitness vs. QTLs | NA | NA | 0.105 (0.047) | 0.199 (0.143) | NA | NA |
| $r^2$, combined models | NA | NA | 0.501 (0.442, 0.561) | 0.577 (0.510, 0.641) | 0.467 (0.411, 0.534) | 0.171 (0.126, 0.235) |
| Added variance combined vs. fitness | NA | NA | 0.087 (0.060) | 0.057 (0.035) | 0.047 (0.021) | 0.044 (0.022) |

DOI: https://doi.org/10.7554/eLife.27167.004

5). In both cases, these QTLs explain 76% of the trait variance. We also found 4 QTLs that significantly affect adaptability, $\Delta X$, at OT and 2 QTLs that affect adaptability at HT. Note that these 2 loci are a subset of the OT adaptability loci (*Supplementary file 5*). Together these adaptability QTLs explain 30% of the total variance in this trait in the OT environment, and 31% in the HT environment, also consistent with our estimates of narrow-sense heritability (*Figure 2* and *Figure 3* and *Table 1*). Due to the limited size of our study we may be missing some QTLs that affect fitness or adaptability. However, the fact that the variance in these traits explained by detected QTLs is similar to our estimates of narrow-sense heritability (*Figure 2*, *Table 1*) indicates that any missing QTLs are unlikely to have large additive effects on the traits.

Interestingly, three of the four adaptability loci were also independently identified as QTLs for founder fitness (*Supplementary file 5*), suggesting that loci that affect founder fitness also affect adaptability. This is consistent with the conclusions of several recent studies, which show that much of the variation in adaptability between closely related microbial strains can be explained by differences in their initial fitness, according to the rule of declining adaptability (*Perfeito et al., 2014*; *Kryazhimskiy et al., 2014*; *Couce and Tenaillon, 2015*). Motivated by these results, we asked whether a similar effect holds in our system as well, even though our founders are several orders of magnitude more diverged than strains analyzed in earlier studies. We found that indeed founder fitness explains much of the variation in adaptability, and follows the rule of declining adaptability (*Figure 3*). Specifically, a model where adaptability declines linearly with founder fitness explains 42% of the total variance in adaptability in the OT environment, and 52% of the adaptability at HT, or about 68% and 80% of the heritable variance, respectively (*Figure 2*). This represents more of the variance explained than the linear QTL models described above, despite the fact that the QTL models have more free parameters. Interestingly, the variance explained by founder fitness is comparable to the narrow-sense heritability, suggesting that founder fitness may be as good of a predictor of adaptability as any linear genetic model.

Since founder fitness explains more variation in adaptability than additive QTL models, and since three of the identified QTLs affect both fitness and adaptability, it is natural to ask whether there are any genetic loci that explain some of the variation in adaptability above and beyond that explained by fitness. Indeed, about 30% and 20% of heritable variation in adaptability remain unexplained by founder fitness in the OT and in the HT environment, respectively (*Figure 2*, *Table 1*). Can we then identify QTLs that explain any of this residual variation?

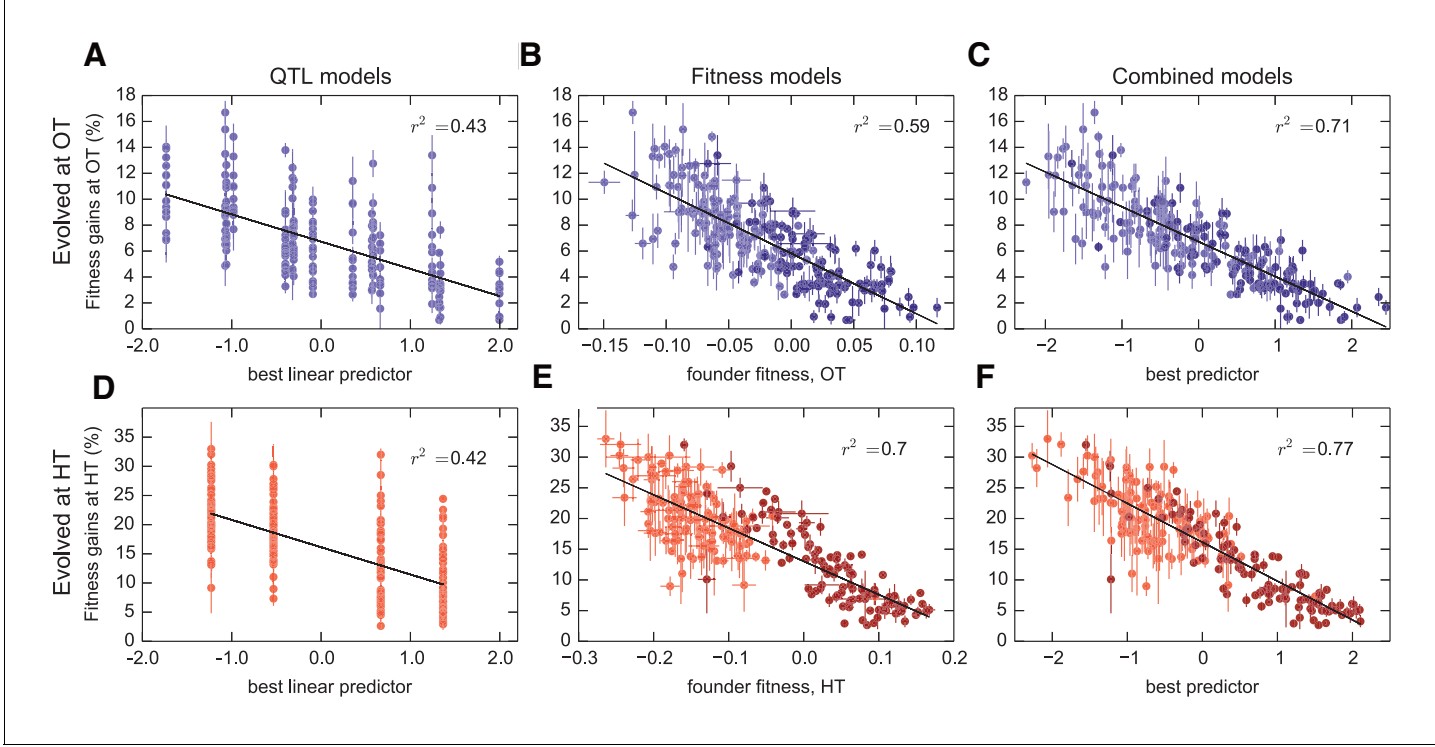

**Figure 3.** Models to predict adaptability. Each point represents the average fitness change of four populations descended from the same founder after 500 generations of evolution in the (**A–C**) OT environment or (**D–F**) HT environment. Error bars denote ±1 s.e.m. Lines show the predictions of different models; $r^2$ indicates the fraction of the variance explained, which is similar to the fraction of the broad-sense heritability explained by the model. (**A,D**) Linear QTL model, where the $x$-axis is the linear combination of identified QTLs that best predicts the fitness gains in each environment. (**B,E**) Model where founder fitness is used as predictor for fitness gain. (**C,F**) Combined model, where the $x$-axis is the linear combination of initial fitness and the identified QTLs that best predicts the fitness gains in each environment. Dark and light shades represent BY and RM alleles at the *KRE33* locus respectively. For model parameters, see *Figure 3—source data 1*.

DOI: https://doi.org/10.7554/eLife.27167.005

The following source data is available for figure 3:

**Source data 1.** Parameters for combined fitness and QTL models for adaptability.
DOI: https://doi.org/10.7554/eLife.27167.006

To address this question, we first asked whether any of the previously identified QTLs (either for founder fitness or for adaptability) can help explain variation in adaptability once we have taken fitness into account by including it as a covariate in the QTL models (Materials and methods). We found that indeed several of these QTLs had a modest effect on adaptability above and beyond fitness (*Supplementary file 5*). The QTL with the largest effect was centered at position 376,315 on chromosome XIV, and we refer to it as the '*KRE33* locus.' This QTL explains 2% percent of residual residual variation in adaptability in the HT environment, and 0.4% percent in the OT environment (*Figure 3*). We previously identified this locus as having the largest effect on founder fitness: segregants with the BY allele at this locus are on average 9% more fit at OT and 19% more fit at HT compared to segregants with the RM allele. After adding pairwise interaction terms between the *KRE33* locus and other detected QTLs, the combined model explains 82% of the heritable variation in adaptability in the HT environment and 88% in the OT environment (*Table 1*). Thus most of the heritable variation in adaptability in our system can be explained by founder fitness, with some additional contributions from genotype at a few adaptability QTL loci. Nonetheless, there remains some residual variation due to further unidentified genotype effects.

To check for strong effect adaptability QTLs not previously identified, we repeated the QTL discovery procedure described above, taking the trait to be the residual variation in adaptability about the regression line in *Figure 3*. We recovered a subset of the already-identified adaptability QTLs, without identifying any further loci.

## Adaptability depends on initial fitness in the 'home' environment

We have seen that the rule of declining adaptability applies in both of our evolution conditions. However, because the initial fitnesses of our founders in the two environments are correlated (*Figure 1*), it is possible that the two patterns of decline in adaptability are not independent. Instead, they both could be driven by some factor common to both environments, but not by the initial fitnesses in those environments. If this were the case, then we would expect that the rate of adaptation of all the strains would be controlled by this factor, and we would not expect a strain with particularly disparate fitnesses in the two environments (i.e. lying away from the trend in *Figure 1B*) to adapt faster in the environment where its fitness was lower. Alternatively, the rule of declining adaptability could apply independently in each environment. In this case, we would expect that a strain with low initial fitness at OT and high initial fitness at HT would adapt rapidly at HT and slowly at OT (and vice versa).

To address this question, we first classify each founder according to how disparate its initial fitness is in HT versus OT (*Figure 4A*). Note that this is the same data as in *Figure 1B* after an appropriate normalization (see Materials and methods), with color representing distance from the diagonal. In *Figure 4B*, we show the average fitness gains of descendants of each of these founders after evolution in each environment. Consistent with a rule of declining adaptability that holds independently in each environment, we find that founders with particularly high initial fitness in HT (green) tend to adapt particularly slowly in HT but not in OT, and vice versa for founders with particularly high initial fitness in OT (purple). In *Figure 4C*, we quantify this trend by plotting the difference in the normalized fitness gains in the two environments as a function of the difference in initial fitnesses in the two environments (see Materials and methods for details). If a common factor drove the patterns of declining adaptability in both environments, then the difference in a founder's initial fitnesses should be uncorrelated with the difference in its adaptabilities in the two environments. In contrast, we see a negative correlation ($r^2 = 0.28$, $p < 10^{-4}$), again consistent with a rule of declining

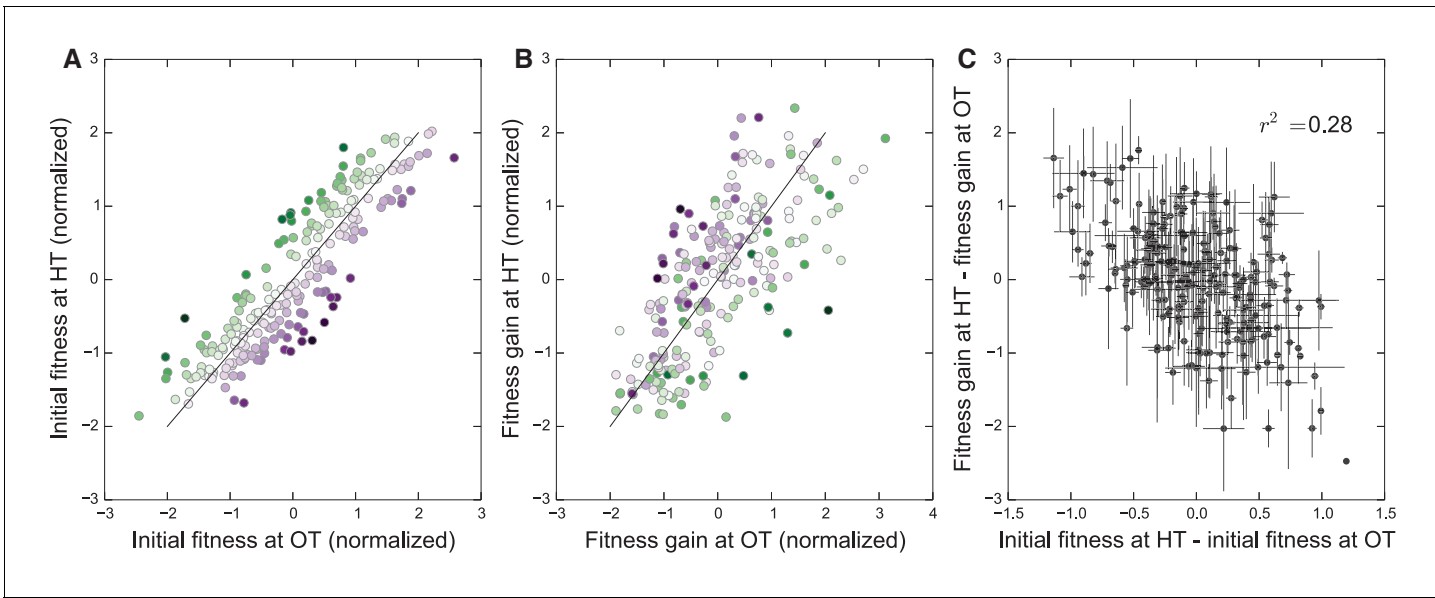

**Figure 4.** Initial founder fitness and adaptability in both environments. (A) Normalized fitness of founders at OT and HT (Materials and methods). Each point is colored according to its deviation from the overall correlation in initial fitnesses. Error bars are omitted for clarity. (B) Normalized fitness increments for all founders (Materials and methods). Colors are the same as in (A); note that points above the diagonal in (A) tend to be found below the diagonal in (B), and vice versa. (C) Difference in fitness increments after adaptation in each environment as a function of the difference in normalized initial founder fitness in HT versus OT. Error bars denote $\pm 1$ s.e.m.

DOI: https://doi.org/10.7554/eLife.27167.007

The following figure supplement is available for figure 4:

**Figure supplement 1.** The effect of fitness in the home environment on adaptability, controlling for the effect of the *KRE33* allele.
DOI: https://doi.org/10.7554/eLife.27167.008

adaptability holding independently in both environments. Thus initial fitness in the home environment is a predictor of its adaptability in that environment, above and beyond a possible common factor that might be driving adaptability in both conditions. We note that these results also hold after controlling for the *KRE33* allele (Materials and methods, *Figure 4—figure supplement 1*).

## The genetic basis of adaptation

Our observations of heritable variation in adaptability indicate that macroscopic epistasis is widespread in our system, and can be largely explained by a model involving founder fitness and a small number of QTLs. However, it is not clear whether this is accompanied by microscopic epistasis between alleles in the founders and potential future mutations. In other words, do different founders accumulate different sets of beneficial mutations as they adapt? And if so, can we identify QTLs or other factors (e.g. founder fitness) that modify the spectrum of mutations that accumulate during adaptation?

To address this question, we sequenced 273 whole-population samples from the final timepoint of all independently evolved populations descended from each of 35 founders (*Supplementary file 3*). We discarded nineteen populations for technical reasons before calling mutations (see Materials and methods), leaving us with a total of 254 sequenced lines (mean 23x depth). We identified SNPs and small indels in this mixed-population sequence data (Materials and methods). We focus here only on mutations that were fixed or estimated to be at high frequency (above 50%) within the population at this final timepoint. We identified a total of 603 mutations (on average 1.3 per OT population and 3.6 per HT population) meeting these criteria. Of these, 449 are nonsynonymous mutations or indels in genes (293 from HT populations, 156 from OT populations).

More fit founders can adapt more slowly because they have access to fewer beneficial mutations, or to beneficial mutations of weaker effect, or both. In any of these cases, we expect that mutations accumulate more slowly over time in populations descended from fitter founders. Consistent with this expectation, we found that the number of mutations identified in a population declines weakly but significantly with the initial founder fitness (*Figure 5*).

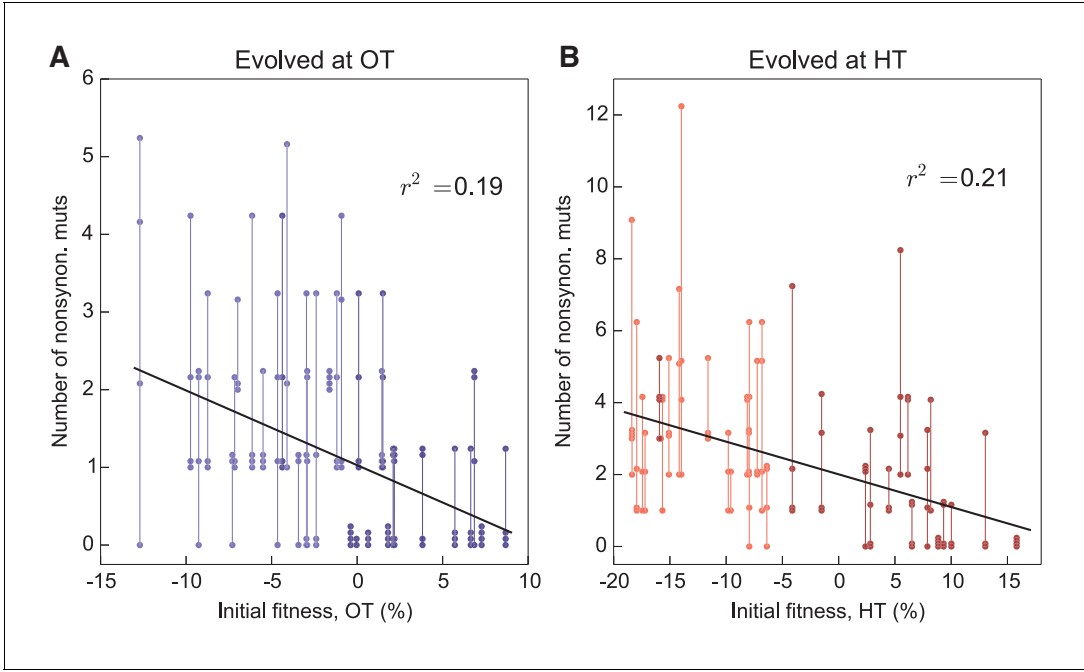

**Figure 5.** Number of fixed nonsynonymous mutations. Each point represents the number of fixed nonsynonymous mutations in a population evolved at (A) OT or (B) HT. Populations descended from the same founder are connected by vertical lines for visual convenience, with founders ordered according to their fitness in the corresponding environment. Dark and light shades represent BY and RM alleles at the *KRE33* locus respectively.
DOI: https://doi.org/10.7554/eLife.27167.009

Beyond variation in the number of mutations, did founder genotype affect the spectrum of mutations available to future adaptation? To address this question we focused on genes that were mutated in multiple populations (genes mutated in a single population are uninformative for this purpose). We found 27 genes in which missense or putative loss of function mutations were called in two or more independent populations (*Figure 6*); these 'multi-hit' genes are expected to be enriched for targets of selection.

The gene with the second-largest number of identified de novo mutations was *KRE33*: 31 independent sequenced populations acquired a mutation in this gene (a total of 12% of our 254 sequenced lines; *Figure 6*). All 31 of these populations are descended from founders that had the less-fit RM allele at the QTL containing the *KRE33* gene (i.e. the same QTL that we identified above as affecting adaptability; *Figure 6*). This observation suggests that genetic variation in the *KRE33* gene itself might have caused the observed variation in adaptability. Since the Kre33 protein is a member of the 90 s preribosomal pathway, we reasoned that the deleterious effect of the RM allele could potentially be compensated not only by mutations in *KRE33* itself, but also by mutations in genes that code for other members of the ribosome biogenesis pathway. Consistent with this hypothesis, we found 108 putatively functional mutations in our sequenced lines in 10 genes (including *KRE33*) that were categorized as belonging to this pathway, all of which arose in descendants of founders that carry the RM allele at the *KRE33* locus (*Figure 6*). Thus, compensation for the deleterious effect of the RM allele in the *KRE33* gene is likely one of the strongest targets of selection in our

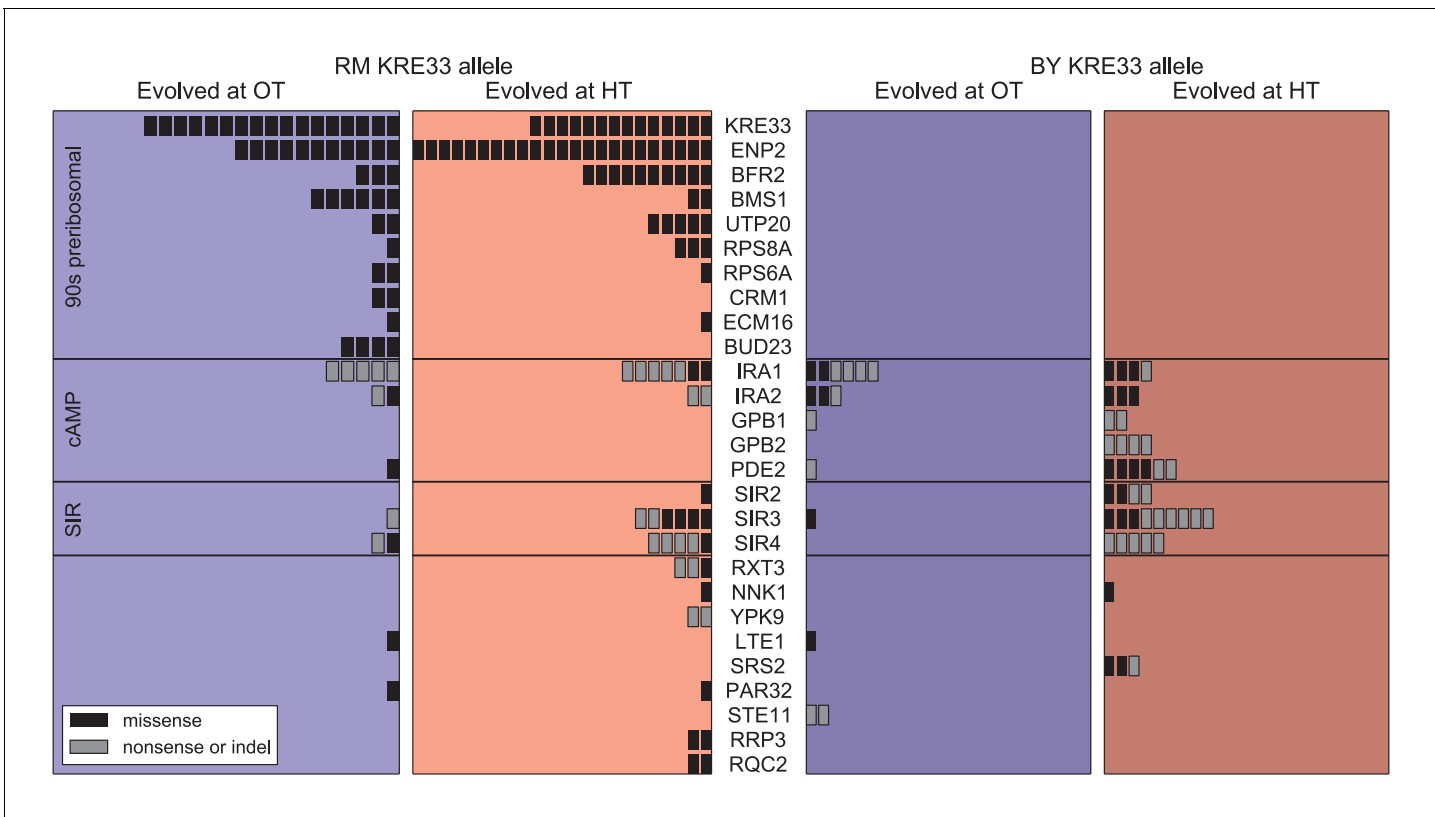

**Figure 6.** Distribution of mutations in multi-hit genes across evolved populations. The number of sequenced populations with a de novo mutation in each multi-hit gene, for each environment and founder *KRE33* identity. Genes are organized by functional groups. For mutual information metrics, see *Figure 6—source data 1*.
DOI: https://doi.org/10.7554/eLife.27167.010

The following source data is available for figure 6:

**Source data 1.** Mutual information between de novo mutations and the *KRE33* allele, the evolution environment, and the genotype.
DOI: https://doi.org/10.7554/eLife.27167.011

experiment, and it likely contributes to the increased adaptability of the founders which carry the RM version of *KRE33*.

Of the remaining 101 mutations in multi-hit genes, 14 out of 15 mutations in *GPB1*, *GPB2* and *PDE2* — genes involved in cAMP regulation — occurred exclusively in populations descended from founders that carry the more-fit BY allele at the *KRE33* locus. This suggests a potential genetic interaction between *KRE33* and this pathway, such that mutations in *GPB1*, *GPB2*, and *PDE2* are more beneficial in the BY *KRE33* background. However, it is also possible that these mutations are always beneficial, but are less likely to fix in the *KRE33*-RM background where they can be lost to clonal interference with more strongly beneficial mutations (e.g. those in the ribosome biogenesis pathway).

The effect of the *KRE33* allele on the genetic basis of future adaptation suggests that other loci that contribute to fitness or adaptability differences between RM and BY strains might also be targets for adaptive de novo mutations. To test this hypothesis, we asked whether multi-hit genes are preferentially found within the 8 QTLs that we identified as influencing founder fitness or adaptability. We found two additional examples of multi-hit genes that lie within the confidence interval for one of these 8 QTLs: *IRA2* and *BFR2* (*Supplementary file 5*). These 2 genes, along with *KRE33*, represent an enrichment over the expected number of genes in common between the set of multi-hit genes and those within QTL confidence intervals (Materials and methods). Additionally, *RAS2*, an essential gene which acts as a master regulator in the cAMP pathway that includes 5 multi-hit genes (including *IRA2* *Figure 6*), is found within one of the QTLs affecting fitness. These results suggest that some loci that underlie variation in fitness among founders are also targets of selection during subsequent evolution. They also suggest that the cAMP pathway, which has been implicated as a target of adaptive de novo mutations in several other experimental evolution studies (*Venkataram et al., 2016*; *Kvitek and Sherlock, 2013*; *Lang et al., 2013*), may also explain some of the fitness differences caused by natural genetic variation in this cross, at least in the specific experimental conditions we analyze here.

With the exception of the cases noted above, which all involve QTLs previously identified as affecting fitness or adaptability, we lack power to identify individual loci in the founder genotype that affect the genetic basis of future adaptation. That is, we cannot find specific QTLs that affect whether mutations in any of the 27 multi-hit genes will be observed in the descendants of a given founder. However, we can still ask whether the mutations that occur during adaptation show a nonrandom association with the founder genotype. We can also ask whether the mutations that occur during adaptation depend on the environment in which evolution was conducted. To do so, we calculated the mutual information $I(G, E; M)$ provided by the founder genotype $G$ and evolution environment $E$ about the set of mutations $M$ that accumulate during adaptation (see Materials and methods for details; note in particular that we control for the identity of the *KRE33* allele throughout this analysis). Intuitively, the mutual information $I(G, E; M)$ measures the amount of information (in bits) that we gain about $M$ by knowing $G$ and $E$. Higher values of $I$ indicate that $G$ and $E$ more strongly predict $M$.

We used $I(G, E; M)$ as a test statistic to determine whether there were nonrandom associations between founder genotype, evolution environment, and the mutations that underlie adaptation. Controlling for the number of mutations acquired by each population, we detected a significant association between the evolution condition and the mutations that accumulate during adaptation. For example, we found putatively functional mutations in *SIR3* and *SIR4* in 34 different populations, 30 of which evolved at HT. However, the overall magnitude of this effect of $E$ on $M$ is small, on the order of 0.1 bits. In addition to this effect of environment, we also find that the genotype $G$ affects $M$. Specifically, controlling for the number of mutations, the *KRE33* allele, and the evolution environment, there is a small additional effect of founder on the identities of de novo mutations. This suggests that while founders with the same allele at the *KRE33* locus draw many common adaptive mutations from the same pool, there are additional founder-specific epistatic interactions.

## The pleiotropic consequences of adaptation

Adaptation to one environment can lead to concomitant fitness gains or losses in other environments, an effect we refer to as pleiotropy for fitness. To characterize these pleiotropic effects in our system, we analyzed how evolution in our OT environment affects fitness in the HT environment, and

vice versa. We can describe the founder fitnesses in both environments as $\vec{W} = (X, Y)$, where $X$ is the initial fitness at OT and $Y$ is the initial fitness at HT. As noted above, the founder fitnesses in the two environments were strongly correlated, especially within each of the two *KRE33* allele groups (*Figure 7A*), suggesting that the alleles that distinguish RM and BY have correlated effects in our environments.

We next calculated the fitness gains for populations adapted to each environment. After evolution in a given environment, a founder with fitness $\vec{W} = (X, Y)$ will have adapted, yielding a descendant line with fitness $\vec{W'} = (X', Y')$. We denote the fitness increments of this population as $\Delta\vec{W} \equiv (\Delta X, \Delta Y) = (X' - X, Y' - Y)$; this vector represents the increase or decrease in fitness of the descendant population in both environments, after evolution in one of the two environments. For clarity, we will use the notation $\Delta_{OT}$ to refer to the fitness increments of descendant populations after evolution in the OT environment, and $\Delta_{HT}$ to refer to the fitness increments after evolution in the HT environment. We refer to the environment in which a given population evolves as its 'home'

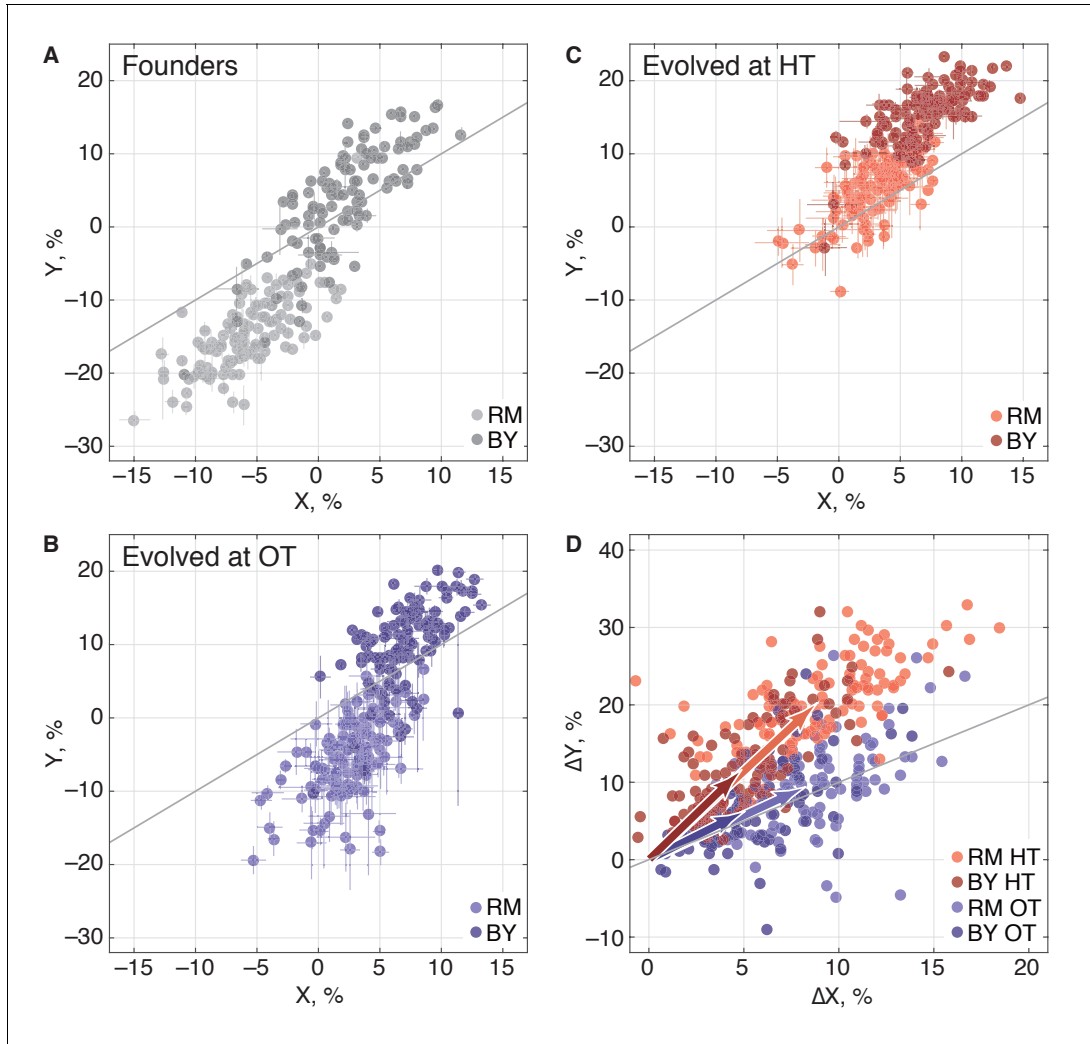

**Figure 7.** Pleiotropic consequences of adaptation. (**A**) The fitness of each founder in the OT environment (*X*) and HT environment (*Y*). Error bars represent ±1 s.e.m. over technical replicates. Dark and light points represent strains carrying the BY and RM alleles at the *KRE33* locus respectively. (**B, C**) Average fitness of all populations descended from each founder. Error bars represent ±1 s.e.m. over replicate populations. Orange and blue colors represent strains evolved at HT and OT, respectively; dark and light shades represent BY and RM alleles at the *KRE33* locus respectively. (**D**) Average fitness increment of all populations descended from each founder; error bars omitted for clarity. Arrows point to the centroids of the respective point clouds.

DOI: https://doi.org/10.7554/eLife.27167.012

environment, and the other environment as its 'away' environment. Thus $\Delta_{OT}X$ and $\Delta_{HT}Y$ are fitness increments in the home environments, and $\Delta_{OT}Y$ and $\Delta_{HT}X$ are fitness increments in the away environments.

We find that, on average, populations adapting to either of our two environments also gained fitness in the other environment (*Table 2*), consistent with our observation that the sets of multi-hit genes in the two environments were highly overlapping. Thus as populations adapt, their fitnesses at OT and HT remain positively correlated (*Figure 7B,C*). Despite this overall pattern of correlated adaptation, we find that the average fitness gain at HT was significantly higher after evolution at HT than after evolution at OT (*Table 2*), as we might expect. However, the converse was not true: populations increased in fitness at OT by about the same amount after evolving at HT as they did after evolving at OT. Because populations evolved at HT and those evolved at OT made almost identical fitness gains at OT, but different gains at HT, evolution in these two conditions must involve different sets of adaptive mutations, despite the overlap in the targets of selection between the two environments. This is consistent with our mutual information analysis above.

## Predicting pleiotropic effects from genotype and founder fitness

We next used the framework of quantitative genetics to analyze the effects of founder genotype on pleiotropy. We find that the pleiotropic consequences of adaptation to our HT environment are highly heritable, with broad-sense heritability $H^2$ of the fitness increases in the away environment, $\Delta_{HT}X \approx 0.62$. This is similar to the heritability of adaptability we found above. In contrast, the pleiotropic consequences of adaptation to our OT environment are much less heritable, with broad-sense heritability of $\Delta_{OT}Y \approx 0.29$, indicating that in this case stochastic evolutionary forces are more important to the outcome than genotype.

As with the adaptability traits analyzed above, we find that the narrow-sense heritability of pleiotropy is substantially lower than the broad-sense heritability (*Figure 8*). Specifically, we find $h^2 \approx 0.48$ for $\Delta_{HT}X$ and $h^2 \approx 0.09$ for $\Delta_{OT}Y$. Thus most of the heritable variation in this trait cannot be explained using an additive QTL model of the underlying founder genotype.

We next sought to understand which factors best explain the heritable variation in pleiotropy. Analogous to our analysis of adaptability, we consider the potential effects of specific QTL loci as well as of founder fitness (now in both home and away environments). We begin by fitting a model in which the fitness increments in the away environment decline linearly with the founder fitness in both the home and the away environments. We find that founder fitness does explain much of the heritable variation in pleiotropy (68% of the heritable variation in $\Delta_{HT}X$ and 43% for $\Delta_{OT}Y$). However, we find that it is only the initial fitness in the *away* environment that is important: once founder fitness in the away environment is included in the model, founder fitness in the home environment adds no further explanatory power, while the converse is not true.

To assess the dependence of pleiotropic outcomes on specific parental alleles, we next looked for QTLs that influence fitness increments in the away environment, over and above founder fitness in that environment. Specifically, we asked whether any of the eight previously-identified QTLs could help predict pleiotropic outcomes over and above founder fitness in the away condition. We find that 6 of these loci, including *KRE33*, do have a significant effect on pleiotropic outcomes (*Supplementary file 5*). Together with founder fitness in the away condition, they explain 75% and 58% of the heritable variation in pleiotropy at OT and HT, respectively (*Figure 8*). To check for strong effect pleiotropy QTLs not previously identified, we also repeated the QTL discovery

**Table 2.** Average fitness increments of populations in home and away environments, in percent. Numbers in parentheses denote 95% confidence intervals.

|  | *Kre33*-RM | | *Kre33*-BY | |
|---|---|---|---|---|
|  | **Evolved at OT** | **Evolved at HT** | **Evolved at OT** | **Evolved at HT** |
| Fitness at OT, % | 8.33 (8.27, 8.38) | 8.86 (8.81, 8.92) | 5.01 (4.94, 5.07) | 4.83 4.76, 4.90 |
| Fitness at HT, % | 9.57 (9.42, 9.77) | 20.33 (20.18, 20.53) | 6.25 (6.12, 6.38) | 11.51 (11.38, 11.64) |

DOI: https://doi.org/10.7554/eLife.27167.013

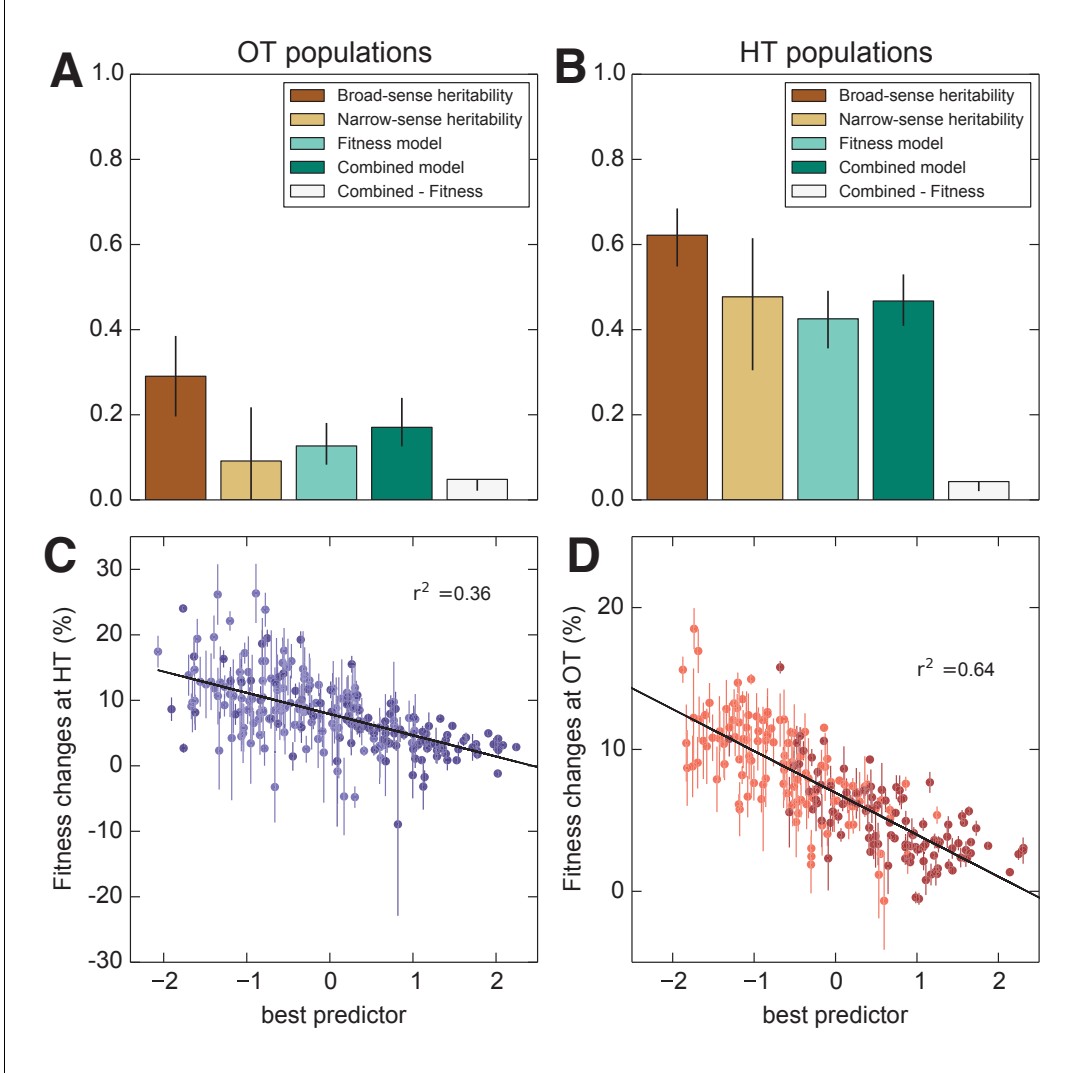

**Figure 8.** Heritability and predictability of pleiotropic fitness gains. (A) Heritability of the fitness gains in the HT environment after evolution at OT, $\Delta_{OT}Y$, and the variance explained by fitness and combined fitness/QTL models. The final (grey) bar indicates the median difference in variance explained between the combined fitness/QTL model and the fitness model under jackknife resampling (Materials and methods). (B) Heritability of $\Delta_{HT}X$ and the variance explained by fitness and combined fitness/QTL models. The final (grey) bar is as in A. (C) Predicted and actual values of $\Delta_{OT}Y$ based on initial fitness at HT and QTL loci (*Supplementary file 5*, column 9). Each point shows the mean over populations descended from the same founder. Error bars denote ±1 s.e.m. over technical replicates (*x*-axis) or over replicate evolved populations (*y*-axis). Dark and light points represent strains carrying the BY and RM alleles at the *KRE33* locus respectively. (D) Predicted and actual values of $\Delta_{HT}X$ based on initial fitness at OT and QTL loci (*Supplementary file 5*, column 10). For model parameters, see *Figure 8—source data 1*.
DOI: https://doi.org/10.7554/eLife.27167.014

The following source data is available for figure 8:

**Source data 1.** Parameters for combined fitness and QTL models for pleiotropy.
DOI: https://doi.org/10.7554/eLife.27167.015

procedure described above, taking the trait to be the residual variation in fitness increments in the away environment after regression against founder fitness in that environment. We recovered a subset of the previously identified QTLs, without identifying any further loci.

# Discussion

In order for evolution to increase or decrease adaptability, there must be genetic variation in this trait that natural selection can act on. Here, we analyzed the extent and genetic basis of variation in adaptability among 230 segregants from a cross between two divergent yeast strains (*Bloom et al., 2013*). By exploiting the structure of the cross, we mapped QTLs that explain observed differences in adaptability. We then compared the predictive power of these additive genetic models with an alternative model where adaptability depends on founder fitness.

Consistent with earlier work, we find that the rate of adaptation is a heritable trait: different founders consistently adapt at different rates. This means that some of the ~50,000 alleles that distinguish BY from RM alter future adaptive trajectories. In principle, these effects could involve just a few founder alleles whose effects on adaptability are independent of each other (e.g., if particular alleles that distinguish RM and BY create opportunities for improvement of independent biological processes). If this were the case, an additive model involving these variants would explain the observed heritable variation in adaptability between founders. However, we find that the best additive QTL model explains at most about half of the observed heritable genetic variation in adaptability.

Instead, we find that a model based on founder fitness explains more of the heritable variation in adaptability, despite having fewer free parameters. This is broadly consistent with earlier work showing that initial fitness explains variation in adaptability among founders that differ only by a few mutations (*Barrick et al., 2010*; *Perfeito et al., 2014*; *Kryazhimskiy et al., 2014*; *Wang et al., 2016*). However, over the larger genetic distances we study here, a substantial fraction of heritable variation in adaptability is not explained by founder fitness. About half of the remaining variation can be explained by including *KRE33* and a small number of other loci as predictors in addition to founder fitness, leaving about 20 percent of the heritable variation unexplained.

We next investigated the pleiotropic consequences of adaptation in one ('home') environment on fitness in another ('away') environment. We found that in the two conditions we considered, adaptation to the home environment leads on average to a fitness gain in the away environment. As with adaptability, these fitness increments in the away environment were heritable. To some extent this is expected: since founder genotype affects adaptability, we expect the founder genotype to affect pleiotropy as well, simply because fitness gains in the home environment are correlated with fitness gains in the away environment. Consistent with this, we find that founder fitness does explain much of the heritable variation in pleiotropy. Surprisingly, however, only initial fitness in the *away* environment is important. These results are consistent with a simple model in which the fitness effects of mutations in any environment are modulated by the fitness of the genetic background in that environment, regardless of how these mutations were acquired. Thus the effects in environment B of mutations accumulated through evolution in environment A are determined primarily by the fitness of the genetic background in environment B.

We also found that the broad-sense heritability of fitness changes in the away environment is asymmetric. For populations evolved at HT, fitness gains in the OT environment were highly heritable, and predicted well by founder fitness in the OT (away) environment. This suggests that the HT populations acquired a set of mutations whose effects at OT are predictable based on the fitness of the genetic background in this condition. Conversely, the populations evolved at OT saw much more variable gains at HT. This implies that the mutations accumulated at OT have effects at HT that are less predictable from founder fitness or genotype.

In addition to founder fitness, we find that the *KRE33* allele and a few other smaller-effect QTLs significantly affect evolutionary outcomes. To some extent, these QTLs affect evolution through their impact on founder fitness. However, five QTLs, including *KRE33*, affect adaptability and in some cases pleiotropy above and beyond their effects on founder fitness. The founder allele at the *KRE33* locus also strongly affects the spectrum of mutations that accumulate during adaptation: founders that start with the less-fit RM allele at *KRE33* accumulate mutations in *KRE33* and in other genes involved in the 90 s preribosomal pathway. These genes are all essential, so these mutations likely involve gain or attenuation rather than loss of function. Surprisingly, despite this effect, *KRE33* still has only a modest impact on adaptability and pleiotropy above and beyond the founder fitness. For example, populations descended from founders with the BY and RM alleles of *KRE33* had similar

patterns of fitness gains in the two environments, despite acquiring different sets of mutations (*Figure 7D*). This suggests that the pleiotropic effects of these two classes of mutations are similar.

Together, these observations suggest the following simple phenomenological picture of adaptation in this system. The average fitness of an adapting yeast population in its home environment is statistically predictable based on the fitness of its founding genotype, as well as the allele at the *KRE33* locus and a few other loci, even for highly divergent strains. Mutations accumulated during adaptation tend to lead to pleiotropic fitness gains in other, related, environments. Our ability to predict these pleiotropic fitness gains depends on the environment: it is strong in one case but weak in the other. This points to the possibility that fitness trade-offs between environments, and other complicating factors, disrupt our ability to predict pleiotropic fitness gains.

We note that our experimental approach has several important limitations. First, we define adaptability as the expected increase in fitness of a line descended from a given founder after 500 generations of evolution in one of two specific and artificial laboratory environments. The effects of founder fitness and genotype on adaptability and pleiotropy might change over different timescales or in a different set of environmental conditions. In particular, the heritability of pleiotropic effects of adaptation may be a consequence of the similarity of the environments we chose, and might be lower across other conditions. Our results also depend on our choice of this specific yeast cross as our source of founder genotypes. In addition, due to the relatively small number of segregants we are able to analyze and the intrinsic noisiness of measurements of adaptability and pleiotropy, our power to identify the genetic basis of these traits is limited. As a result, we have likely missed some QTLs affecting adaptability and pleiotropy (though our estimates of narrow-sense heritability do suggest that any potentially missing QTLs have minor additive effects on these traits). Further work across other environments and with other strains and organisms will be needed to test the generality of our conclusions. However, despite these important caveats, our results provide additional support to a growing body of evidence that a rule of declining adaptability may be general feature of microbial evolution.

Our results are consistent with earlier studies showing that microscopic epistasis is common among mutations that accumulate along the line of descent in laboratory microbial evolution experiments. However, it has been unclear whether this microscopic epistasis leads to macroscopic epistasis, in which related genotypes have different distributions of fitness effects of new mutations. In principle, many patterns of microscopic epistasis could be consistent with the same pattern of macroscopic epistasis, or with no macroscopic epistasis at all. In turn, many types of macroscopic epistasis could be consistent with the same pattern of variation in adaptability, since the expected rate of adaptation depends in a complex way on the entire distribution of fitness effects of new mutations and on population parameters. For example, it is possible to have extensive macroscopic epistasis without any variation in adaptability.

One way to study differences in adaptability would be to take a bottom-up approach, measuring patterns of microscopic epistasis and using these to characterize macroscopic epistasis and then adaptability. We have instead taken the opposite approach, by directly measuring genetic variation in adaptability. We have shown that even across large genetic distances, less-fit initial genotypes adapt more rapidly than more-fit initial genotypes. Our results cannot explain what type of microscopic epistasis, if any, underlies this pattern of declining adaptability. Previous work indicates that some adaptive mutations exhibit negative ('diminishing returns') epistasis, which is implicated in driving the pattern of declining adaptability (*Chou et al., 2011*; *Khan et al., 2011*; *Kryazhimskiy et al., 2014*; *Wünsche et al., 2017*), but we do not have direct evidence for or against this type of microscopic epistasis in the present experiment. However, we do find at least one significant microscopic epistatic interaction that affects adaptability at both fitness and genotype levels in our HT environment, which involves the allele at the *KRE33* locus. This microscopic epistasis also exists in our OT environment, as evidenced by its effect on the genetic basis of adaptation, and on the pleiotropic consequences in the HT condition. Surprisingly, however, this microscopic epistasis has a very weak effect on the rate of adaptation in the OT environment, above the effect we expect from lower founder fitness. This is an example of a case where idiosyncratic microscopic epistasis does not lead to differences in adaptability, either because the microscopic epistasis does not lead to macroscopic epistasis, or because it leads to macroscopic epistasis which does not significantly affect adaptability.

## Materials and methods

### Founders of evolving populations

We selected the founders of our evolving populations from a panel of haploid segregants constructed by *Bloom et al. (2013)*. Briefly, Bloom *et al.* mated strains derived from RM11-1a and BY4716, sporulated the resulting diploids, and then isolated and sequenced 1000 haploid offspring (*Bloom et al., 2013*). We used the first 230 MATa segregants from this cross as our founder strains (*Supplementary file 1*).

### Experimental evolution

We established eight populations from each founder. To avoid artifacts arising from shared standing variation, we founded each population from an independent colony. We propagated each of the resulting 1840 lines for 500 generations in batch culture in unshaken flat bottom polypropylene 96-well plates. We maintained half of the lines (four descended from each founder) in 128 $\mu$L of rich laboratory media, YPD (1% Bacto yeast extract (VWR #90000–722), 2% Bacto peptone (VWR #90000–368), 2% dextrose (VWR #90000–904)) at 30°C with daily $1 : 2^{10}$ dilutions (the OT environment). We maintained the other half of the lines in 128 $\mu$L of synthetic complete media (0.67% YNB with nitrogen (Sunrise Science #1501–250), 0.2% SC (Sunrise Science # 1300–030), 2% dextrose) at 37°C with daily $1 : 2^9$ dilutions (the HT environment). All liquid handling was conducted using a BiomekFX robot (Beckman Coulter). Prior to dilution, cultures were resuspended by shaking at 1200 rpm for 2 min on a Titramax 100 plate shaker.

As previously described by *Lang et al. (2011)*, this protocol results in approximately ten generations per day (for the OT environment) or nine generations per day (for the HT environment) at an effective population size of $N_e \approx 10^5$. Every 7 days, aliquots from each population were mixed with glycerol to 25% and kept at -80°C for long-term storage. To check for cross-contamination, each plate contained a unique pattern of blank wells. No cross contamination events were observed during the evolution. However, some wells were lost due to pipetting artifacts. These wells were excluded from all analysis, leaving a total of 910 evolved lines in the OT environment and 839 in the HT environment.

### Fitness assays

We conducted fitness assays as described previously (*Kryazhimskiy et al., 2014*). Briefly, we measured fitness by competing founding clones and evolved populations against a common reference strain. To construct the reference, we selected a segregant with intermediate fitness from the initial RMxBY cross (segregant LK3-B08; see *Supplementary file 1* for the genotype). We integrated an mCitrine-KanMX cassette at the *ho* locus of this segregant. The marker was obtained via a digest of plasmid pEJ03-mCitrine-KanMX-HO (*Supplementary file 4*) with pMEI and transformed using standard yeast genetic techniques (*Adams et al., 1998*). Transformants were selected based on growth on G418, and the Citrine + phenotype was confirmed via flow cytometry.

For each fitness assay, we first allowed both the evolved and reference strains to acclimatize in the relevant environment for 24 hr. We then mixed these strains in approximately equal proportions and propagated them for two days using the same protocol as for evolution. We used Fortessa and LSRII flow cytometers (BD Biosciences) to count the ratio of evolved and reference strains 1 and 2 days (approximately 10 generations and 20 generations respectively, or 9 and 18 in the HT environment), counting approximately 30,000 cells in each measurement. We estimated the fitness of the evolved strain relative to the reference as $s = \frac{1}{\tau} \ln(\frac{n_{e,f}}{n_{r,f}} \frac{n_{r,i}}{n_{e,i}})$, where $\tau$ is the time between measurements in generations, $n_{e,i}, n_{e,f}$ are the initial and final counts of the evolved strain, and $n_{r,i}, n_{r,f}$ are the initial and final counts for the reference. In the HT environment (synthetic medium), we found that in a sample of pure reference cells, 98.5% were fluorescent. To account for this, we adjusted the fitness estimates for this environment slightly: $s = \frac{1}{\tau} \ln(\frac{n_{e,f} - p n_{r,f}}{n_{r,f}} \frac{n_{r,i}}{n_{e,i} - p n_{r,i}})$, where $p = 0.015$ is the non-fluorescent proportion of reference cells. To estimate the error on the founder fitnesses, we measured the fitnesses of the eight founder clones picked from each segregant in their home environment (i.e. four at OT and four at HT). We measured the fitnesses of each final evolved population at both OT and HT. To obtain estimates of the technical error for these measurements, we chose 24 final populations

at random and made 8 technical replicate measurements of each in each of the two measurement conditions.

## Sequencing and mutation calling

We sequenced all 273 populations descended from 35 founders in both environments (this excludes 7 populations descended from these founders that were lost due to pipetting artifacts during evolution, as described above). The founders selected for this sequencing were chosen to ensure approximately equal representation of each parental *KRE33* allele (see below), but were otherwise random. To focus on common mutations within each population, we sequenced mixed whole-population samples. We prepared libraries for sequencing as described previously (*Baym et al., 2015*), and performed whole-genome sequencing on an Illumina HiSeq 2500 in rapid run mode. Fastq files have been deposited with the NIH SRA (Sequence Read Archive), under accession number SRP102877.

To process the whole-population sequence data, we first trimmed reads using Trimmomatic function 'Illuminaclip', with options Leading:20, Trailing:20 (*Bolger et al., 2014*). We merged overlapping paired-end reads using the bbmerge function of bbtools (v36.77). We then used Breseq v0.27.1 (*Deatherage and Barrick, 2014*) with option '-p' (polymorphic mode) to align reads to the S288c public reference genome (version R64-2-1, downloaded 13 January 2015 (*Engel et al., 2014*)). We did not directly use the SNP and indel calls made by Breseq. Instead, we used the Samtools mpileup function and a custom python script to create an unfiltered list of the calls in each population at each base pair. To call de novo mutations, we first filtered out fixed differences between the parents by removing all sites where the alternate allele is in the majority in at least 6 lines. We then filter error-prone sites by removing sites where the alternate allele is above 10% frequency in 5 or more populations. These errors typically arise from alignment artifacts. We called a mutation in a population when it occurred in more than 50% of the reads, with support for the alternate allele in at least 4 reads. Although we are sequencing mixed-population samples, 57% of the mutations that we called were supported by 100% of the reads for that population, and 68% were at $\geq$90% frequency.

We discarded 12 sequenced populations due to insufficient coverage and/or library prep failure. We also checked the genotypes of our founding clones at loci where the RM and BY parents differ, to verify that they matched the genotypes reported by *Bloom et al. (2013)*. We found that 254/261 (97%) of genotypes matched those reported by *Bloom et al., (2013)*, but 7 (3%) were incorrect. We attribute this to errors in picking clones to set up the founders for evolution. We discarded these 7 populations from the sequence data analysis.

We detected 3 SNVs that were shared among multiple descendant populations of the same founder (7 populations in total). For two of these cases, the SNV appears in one OT population and one HT population; in the third case, it appears in one OT population and two HT populations. These mutations likely arose prior to picking clones to found populations, and we discarded them from the analysis.

## Broad-sense heritability

Broad-sense heritability $H^2$ is the fraction of observed variance in a phenotype that can be attributed to genetic differences between founders. We find the broad-sense heritability of founder fitness, $H^2_X$, by partitioning the total observed variance in fitness, $\sigma^2_{X,\mathrm{t}}$, into the component $\sigma^2_{X,\epsilon}$ arising from measurement error and the component $\sigma^2_{X,\mathrm{f}}$ arising from founder genotype. The broad-sense heritability of initial fitness is then

$$H^2_X = \frac{\sigma^2_{X,\mathrm{f}}}{\sigma^2_{X,\mathrm{t}}}.$$

We estimate the components of variance according to the formulas

$$\sigma^2_{X,\mathrm{t}} = \frac{1}{n_g}\sum_{i=1}^{n_g}\left(X_i - \overline{X}_{..}\right)^2, \quad \sigma^2_{X,\epsilon} = \frac{1}{n_g}\sum_{i=1}^{n_g}\sigma^2_{X,\epsilon,i}, \quad \sigma^2_{X,\mathrm{f}} = \sigma^2_{X,\mathrm{t}} - \sigma^2_{X,\epsilon}. \tag{1}$$

Here $n_g = 230$ is the number of founders and we denote the replicate fitness measurement $k$ of founder $i$ by $X_{ik}$. We have also defined $X_i \equiv \overline{X}_{i\cdot} = \frac{1}{n_{r,i}}\sum_{k=1}^{n_{r,i}} X_{ik}$ as the estimate of fitness of founder $i$ is

the unbiased estimate of error variance in the fitness measurement of founder $i$ is the number of technical replicate measurements for founder $i$; and $\overline{X}_{..} = \frac{1}{n_g}\sum_{i=1}^{n_g} X_i$ is the mean fitness across all founders. We calculate confidence intervals for $H^2$ using a leave-$\frac{n}{2}$-out jackknife on genotypes.

Similarly, to estimate the broad-sense heritability of the fitness increment after evolution, $H^2_{\Delta X}$, we partition the total observed variance in fitness increment, $\sigma^2_{\Delta X,\mathrm{t}}$, into the component due to measurement error plus evolutionary stochasticity (which arises due to random variation in fitness gains between populations descended from the same founder), $\sigma^2_{\Delta X,\mathrm{p}}$, and the component $\sigma^2_{\Delta X,\mathrm{f}}$ due to systematically different fitness gains in populations descended from different founders. The broad-sense heritability in fitness increment is then

$$H^2_{\Delta X} = \frac{\sigma^2_{\Delta X,\mathrm{f}}}{\sigma^2_{\Delta X,\mathrm{t}}}.$$

We estimate the variance components as follows. The estimate of the fitness increment of population $j$ descended from founder $i$ is $\Delta X_{ij} = X'_{ij} - X_i$, where $X_i$ is the estimate of fitness of founder $i$ given above and $X'_{ij}$ is estimate of final fitness of the focal population. We estimate the variance $\sigma^2_{\Delta X,\mathrm{p}}$ due to measurement error and evolutionary stochasticity as

$$\sigma^2_{\Delta X,\mathrm{p}} = \frac{1}{n}\sum_{i=1}^{n_g} \frac{n_{p,i}}{n_{p,i}-1} \sum_{j=1}^{n_{p,i}} (\Delta X_{ij} - \overline{\Delta X_{i\cdot}})^2,$$

where $\overline{\Delta X_{i\cdot}} = \frac{1}{n_{p,i}}\sum_{j=1}^{n_{p,i}} \Delta X_{ij}$ is the mean fitness increment in populations descended from founder $i$, and $n$ is the total number of descendant populations. We estimate the total variance in fitness increment as

$$\sigma^2_{\Delta X,\mathrm{t}} = \frac{1}{n}\sum_{i=1}^{n_g}\sum_{j=1}^{n_{p,i}} (\Delta X_{ij} - \overline{\Delta X_{..}})^2,$$

where $\overline{\Delta X_{..}} = \frac{1}{n}\sum_{i=1}^{n_g}\sum_{j=1}^{n_{p,i}} \Delta X_{ij}$ is the mean fitness increment across all populations. Finally, the variance due to founder genotype is

$$\sigma^2_{\Delta X,\mathrm{f}} = \sigma^2_{\Delta X,\mathrm{t}} - \sigma^2_{\Delta X,\mathrm{p}}.$$

As above, we calculate confidence intervals on the estimate of $H^2_{\Delta X}$ using a leave-$\frac{n}{2}$-out jackknife on genotypes.

Data analysis for this and subsequent sections was performed in Python v. 2.7 using custom scripts, available at: https://github.com/erjerison/adaptability (*Jerison and Kryazhimskiy, 2017*; a copy is archived at https://github.com/elifesciences-publications/adaptability ).

## Narrow-sense heritability

We estimate the narrow-sense heritability of two types of traits: the fitness of founding genotypes, $X_i$, and the fitness increment of population $ij$, $\Delta X_{ij}$. Let the trait value of a particular individual be $Y_i$. Without perfect knowledge of which loci affect the trait, we can still estimate the narrow-sense heritability based on how the covariance in trait values between individuals depends on their genetic relatednesses (*Yang et al., 2010*; *Zuk et al., 2012*).

Supposing the trait $Y_i$ was a linear combination of contributions from all loci, we have

$$Y_i = \alpha + \sum_{k=1}^{m} g_{ik} a_k + \epsilon_i, \tag{2}$$

where $g_{ik} = -\frac{p}{\sqrt{mp(1-p)}}$ if individual $i$ carries the RM allele at locus $k$ or $g_{ik} = \frac{1-p}{\sqrt{mp(1-p)}}$ if it carries the BY allele at locus $k$, $p$ is the frequency of the BY allele at locus $k$, and $m$ is the total number of loci. We denote the contribution of allele $k$ to the trait as $a_k$. If we take this to be a random effects model, with $a \sim N(0, \sigma_a^2)$ and $\epsilon \sim N(0, \sigma_\epsilon^2)$, then the variance-covariance matrix $\mathbf{V}$ of $Y_i$ is $\mathbf{V} = \mathbf{R}\sigma_a^2 + \mathbf{I}\sigma_\epsilon^2$ (*Yang et al., 2010*), where $R_{ij} = \sum_k g_{ik} g_{jk}$ is the relatedness between two segregants.

The narrow-sense heritability is then $h^2 = \frac{\sigma_a^2}{Var(Y)}$, the ratio of the additive variance to the total phenotypic variance.

We fit $\sigma_a^2$ using standard REML optimization, and estimated $\hat{h}^2 = \frac{\hat{\sigma}_a^2}{Var(Y)}$. To maintain consistency, we calculated confidence intervals using a leave-$\frac{n}{2}$-out jacknife on segregants.

## Mapping QTLs

Following *Bloom et al. (2013)*, we took an iterative approach to identify QTLs for a trait. As before, let $Y_i$ be the phenotypic trait value for founder $i$. At each iteration we detect one QTL, so that after completing iteration $k$, we will have identified $k$ QTLs. At iteration $k + 1$, we first fit the linear model

$$Y_i = \alpha^{(k)} + \sum_{\ell=1}^{k} g_{i\ell} a_\ell^{(k)} + \epsilon_i^{(k)}. \tag{3}$$

Here $\alpha^{(k)}$ and all $a_\ell^{(k)}$ are fitting parameters and $\epsilon_i^{(k)}$ are noise terms, which are normally distributed random variables with mean 0 and variance $\sigma_{\epsilon,k}^2$. As before, $g_{i\ell}$ denotes the genotype of founder $i$ at locus $\ell$. Note that, at the first iteration, the second term in *equation (3)* is absent. We then calculate the residuals $y_i^{(k+1)} = Y_i - \left( \hat{\alpha}^{(k)} + \sum_{\ell=1}^{k} g_{i\ell} \hat{a}_\ell^{(k)} \right)$, where the hat symbol denotes the fitted parameters. Next, for each locus $\ell$, we calculate the log of the odds (LOD) score as $-n_g/2 \log_{10}(1 - r_{k,\ell}^2)$, where $r_{k,\ell}^2$ is the Pearson correlation coefficient between the allele at locus $\ell$ and $y_i^{(k+1)}$, and $n_g$ is the number of founders. For this correlation analysis, we weigh observations by the square root of the number of technical replicate measurements. Next, we perform a permutation test (by permuting $y_i^{(k+1)}$ values) to determine whether the QTL with the largest value of the test statistic is significant at the 0.05 level. If it is, it becomes the $k + 1$th QTL, and we proceed to the next iteration. If it is not, QTL detection is terminated. At each round, we calculate a confidence interval for the location of the QTL based on an LOD decline of 1.5.

We carry out this QTL detection procedure separately for two traits, initial fitness $X$ and fitness increment $\Delta X$, in each of the two environments, OT and HT.

## Fitness models for adaptability

We model the dependence of adaptability on fitness using the standard linear regression,

$$\Delta X_i = \alpha + \beta X_i + \epsilon_i, \tag{4}$$

where $\Delta X_i = \frac{1}{n_i} \sum_{j=1}^{n_i} X'_{ij} - X_i$ is the mean fitness increment of founder $i$, and $X_i$ is the fitness of founder $i$.

Our null model is that the true fitness gains are independent of the founder fitnesses. However, our estimates of $\Delta X_i$ also include measurement error, and these errors are not independent of the measurement errors in $X_i$. These errors could generate spurious correlations between $\Delta X_i$ and $X_i$. Note that we expect this effect to be small, because the error variance in $X_i$ is 0.5% of the total variance in $X_i$ at OT and 0.8% at HT. To assess the influence that these spurious correlations could have had on our results, we also fit the model

$$\Delta X_i = \alpha + \beta \tilde{X}_i + \epsilon_i, \tag{5}$$

where $\tilde{X}_i = X_i + \delta_i$, $\delta_i \sim N(\sigma_{X,\epsilon,i}^2, 0)$, where $\sigma_{X,\epsilon,i}^2$ is the error variance in founder fitness $i$, as defined above. This additional error lowers the correlation from 0.522 to 0.518 at HT and 0.417 to 0.415 at OT.

To assess the extent to which fitness in either environment predicts fitness changes in the other condition, we fit the model

$$\Delta X_{i,a} = \alpha + \beta X_i + \gamma Y_i + \epsilon_i, \tag{6}$$

where $\Delta X_{i,a}$ is the fitness increment of population $ij$ in the 'away' environment, in which it did not

evolve, $X_i$ is the founder fitness in the away environment, and $Y_i$ is the founder fitness in the home environment. As above, to control for spurious correlations due to experimental noise, we also fit

$$\Delta X_{ij,a} = \alpha + \beta \tilde{X}_i + \gamma Y_i + \epsilon_i. \tag{7}$$

We assessed the significance of each coefficient in both models.

## Combined QTL and fitness model

To determine whether any loci influence adaptability over and above fitness, we created a master list of all distinct QTLs detected for the initial fitness or fitness increment as traits. We tested whether each of these loci significantly improved predictions of delta fitness in both environments. We also included terms for potential pairwise interactions between the largest-effect QTL (*KRE33*) and the others. Specifically, we fit the model

$$\Delta X_i = \alpha + \beta X_i + \gamma g^* + \sum_{\ell=1}^{k} g_{i\ell} a_\ell + \sum_{\ell=1}^{k} g^* g_{i\ell} b_\ell + \epsilon_i, \tag{8}$$

where $g^*$ is an indicator variable on the *KRE33* allele, taking values $0.5$ or $-0.5$; the $g_{i\ell}$ are indicator variables on the previously identified QTLs, also taking values $0.5$ or $-0.5$, and $\alpha$, $\beta$, $\gamma$, $a_\ell$, and $b_\ell$ are coefficients.

We fit this model for fitness increment in each environment separately. In each case, we evaluated the significance of the coefficients using a Bonferroni-corrected F-test. We refit the model including only terms with significant coefficients to determine the best predictor of delta fitness in each environment (*Figure 3*, *Figure 3—source data 1*).

To check for additional loci that influenced adaptability over and above fitness, we repeated the QTL detection procedure described in the previous section, taking the trait to be residual differences in adaptability beyond those explained by initial fitness. We did not find any additional QTLs beyond those previously identified.

We repeated this analysis for fitness increment in the alternative condition (*Figure 8*, *Figure 8—source data 1*).

## Testing the effect of fitness in the home environment on adaptability

To establish whether the pattern of declining adaptability in our two environments is entirely driven by a common factor, or whether the fitness in the home environment adds additional predictive power, we tested whether the difference in initial fitness between environments of a particular founder was predictive of the difference in fitness increments after adaptation to each environment. To do so, we define the normalized fitness of founder $i$ at OT as

$$\tilde{X}_i = \frac{X_i - \overline{X}}{\mathrm{Std}(X)}, \tag{9}$$

where $X_i$ is the fitness of founder $i$ at OT, $\overline{X}$ is the average fitness of all founders at OT, and $\mathrm{Std}(X)$ is the standard deviation of all founder fitnesses at OT. We analogously define $\tilde{Y}_i$ to be the normalized fitness of founder $i$ at HT. We define the normalized fitness increment of founder $i$ at OT as

$$\Delta \tilde{X}_i = \frac{\Delta X_i - \overline{\Delta X}}{\mathrm{Std}(\Delta X)}, \tag{10}$$

where $\Delta X_i$ is the average fitness increment of populations descended from founder $i$ at OT, $\overline{\Delta X}$ is the average fitness increment at OT over all founders, and $\mathrm{Std}(\Delta X)$ is the standard deviation of the fitness increment of all founders at OT. We analogously define $\Delta \tilde{Y}_i$ to be the normalized fitness increment of founder $i$ at HT. Note that the normalization by the standard deviation is necessary to meaningfully subtract these quantities, because fitnesses in OT and HT are on different scales (as seen in *Figure 1B*, fitness differences between a typical pair of segregants are almost twice as large in HT as in OT).

In *Figure 4A*, we plot $\tilde{X}_i$ versus $\tilde{Y}_i$, and the line $y = x$. The points are colored by $\tilde{X}_i - \tilde{Y}_i$, which is proportional to the distance to the diagonal. In *Figure 4B*, we plot $\Delta \tilde{X}_i$ versus $\Delta \tilde{Y}_i$, and the line $y = x$,

with the colors as in *Figure 4A*. In *Figure 4C*, we plot $\tilde{X}_i - \tilde{Y}_i$ versus $\Delta\tilde{X}_i - \Delta\tilde{Y}_i$. We calculated the Pearson correlation coefficient for $\tilde{X}_i - \tilde{Y}_i$ versus $\Delta\tilde{X}_i - \Delta\tilde{Y}_i$, and tested significance by bootstrapping over segregants. We also repeated this analysis controlling for the effect of the *KRE33* allele. Specifically, we defined the same quantities as above, with means and standard deviations taken over segregants with the RM and BY *KRE33* alleles separately (*Figure 4—figure supplement 1*).

## Mutual information analysis

To detect associations between the properties of a population and the de novo mutations that occur during adaptation, we used a test statistic based on mutual information. For clarity, we will first define the mutual information between a property and mutations in one gene, and then describe how we combine the information from all the genes. Let $W$ be the property, with groups $W_1, \ldots W_n$ (for example, if the property is the evolution condition, then we have $W_1 = \mathrm{HT}$, $W_2 = \mathrm{OT}$). Let our gene of interest be $g_l$, and let $m$ be an indicator variable with value 1 when a de novo mutation was called in a population in this gene, 0 otherwise. Then the mutual information between the property $W$ and gene $g_l$ is

$$I(W, g_l) = \sum_{W=(W_1, \ldots W_n)} p(W) \sum_{m=(0,1)} p(m|W) \log_2 \frac{p(m|W)}{p(m)}, \tag{11}$$

where we estimate the probabilities based on observed counts: $p(m)$ is the frequency of observing a mutation in $g_l$ across all populations, $p(m|W=W_j)$ is the frequency of observing the mutation among populations with property $W_j$, and $p(W=W_j)$ is the proportion of populations with property $W_j$.

We will also often be interested in the mutual information conditional on a second property $Z$. This will allow us to determine whether a property carries additional information after we have already taken a known predictor into account. This can be calculated as

$$I(W, g_l | Z) = \sum_{Z=(Z_1, \ldots Z_q)} p(Z) \sum_{W=(W_1, \ldots W_n)} p(W|Z) \sum_{m=(0,1)} p(m|W,Z) \log_2 \frac{p(m|W,Z)}{p(m|Z)}. \tag{12}$$

As our test statistic, we take the sum of the mutual information across all of the genes

$$M(W|Z) = \sum_{g_l} I(W, g_l | Z), \tag{13}$$

where for generality we have included the possibility of conditioning on one or more factors. Let $W = Kr$ be a population's *KRE33* allele, which can be either RM or BY. Let $W = E$ be the evolution environment, and let $W = F$ be the population's founding genotype. Then we calculate $M(Kre)$, $M(E|Kre)$, and $M(F|Kre,E)$.

We calculate the null distributions of these statistics by permuting mutations among populations, holding the number of mutations per population fixed. We report the mutual information as $M(Kre) - \bar{M}_p(Kre)$, where $\bar{M}_p(Kre)$ is the mean of the null distribution; $M(E|Kre) - \bar{M}_p(E|Kre)$, and $M(F|Kre,E) - \bar{M}_p(F|Kre,E)$, with 95% confidence intervals calculated from the null distributions (*Figure 6—source data 1*).

## Comparing QTL locations and de novo mutations

To ask whether we see multi-hit mutations near QTL loci more often than we would expect, we determined whether there was an enrichment of common members between the list of 27 multi-hit genes and the list of 99 genes within QTL confidence intervals. Because we excluded dubious ORFs from both lists, we took the number of genes in the yeast genome to be 5858. We computed the expected number of common members to be 0.95.

## Acknowledgements

We thank Benjamin Good and Andrew Murray for helpful comments and suggestions, and Alex Nguyen Ba for technical help. This work was supported by a National Science Foundation Graduate Research Fellowship (to ERJ), a Burroughs Wellcome Career Award at the Scientific Interface (to SK), by National Institutes of Health Grant R01 GM102308 and the Howard Hughes Medical Institute

(LK), and by the Simons Foundation (grant 376196), grant PHY 1313638 from the NSF, and grant R01 GM104239 from the NIH (MMD). Computational work in this study was performed on the Odyssey cluster supported by the Research Computing Group at Harvard University.

## Additional information

### Competing interests
Leonid Kruglyak: Reviewing editor, *eLife*. The other authors declare that no competing interests exist.

### Funding

| Funder | Grant reference number | Author |
| --- | --- | --- |
| National Institutes of Health | R01GM102308 | Leonid Kruglyak |
| Simons Foundation | 376196 | Michael M Desai |
| National Science Foundation | PHY 1313638 | Michael M Desai |
| Howard Hughes Medical Institute | Investigator | Leonid Kruglyak |
| National Institutes of Health | R01GM104239 | Michael M Desai |
| National Science Foundation | Graduate Research Fellowship | Elizabeth R Jerison |
| Burroughs Wellcome Fund | Career Award at the Scientific Interface | Sergey Kryazhimskiy |

The funders had no role in study design, data collection and interpretation, or the decision to submit the work for publication.

### Author contributions
Elizabeth R Jerison, Sergey Kryazhimskiy, Conceptualization, Software, Formal analysis, Investigation, Methodology, Writing—original draft, Writing—review and editing; James Kameron Mitchell, Investigation, Data acquisition and analysis; Joshua S Bloom, Formal analysis, Methodology, Writing—review and editing; Leonid Kruglyak, Investigation, Methodology, Writing—review and editing; Michael M Desai, Conceptualization, Formal analysis, Funding acquisition, Writing—original draft, Writing—review and editing

### Author ORCIDs
Elizabeth R Jerison  http://orcid.org/0000-0003-3793-8839
Sergey Kryazhimskiy  http://orcid.org/0000-0001-9128-8705
Leonid Kruglyak  http://orcid.org/0000-0002-8065-3057
Michael M Desai  http://orcid.org/0000-0002-9581-1150

### Decision letter and Author response
Decision letter https://doi.org/10.7554/eLife.27167.024
Author response https://doi.org/10.7554/eLife.27167.025

## Additional files

### Supplementary files
• Supplementary file 1. Genotypes of segregants used in this study. Note this data is adapted from *Bloom et al. (2013)*, with marker loci that are redundant for this subset of the segregant panel removed.
DOI: https://doi.org/10.7554/eLife.27167.016

• Supplementary file 2. Fitness measurements in the OT and HT environments for each founder and descendant population.
DOI: https://doi.org/10.7554/eLife.27167.017

• Supplementary file 3. All mutations called at frequencies of $\geq 50\%$ in evolved populations.
DOI: https://doi.org/10.7554/eLife.27167.018

• Supplementary file 4. Annotated sequence for pEJ03-mCitrine-KanMX-HO.
DOI: https://doi.org/10.7554/eLife.27167.019

• Supplementary file 5. QTL locations, and genes within confidence intervals, for all traits.
DOI: https://doi.org/10.7554/eLife.27167.020

• Transparent reporting form
DOI: https://doi.org/10.7554/eLife.27167.021

## Major datasets

The following dataset was generated:

| Author(s) | Year | Dataset title | Dataset URL | Database, license, and accessibility information |
|---|---|---|---|---|
| Jerison ER, Kryazhimskiy S, Mitchell J, Bloom JS, Kruglyak L, Desai MM | 2017 | Data from: Genetic variation in adaptability and pleiotropy in budding yeast | https://www.ncbi.nlm.nih.gov/sra/?term=SRP102877 | Publicly available at the NCBI Sequence Read Archive (accession no. SRP102877) |

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
