## [Decision Letter]

Thank you for submitting your article "Genetic variation in adaptability and pleiotropy in budding yeast" for consideration by *eLife*. Your article has been reviewed by three peer reviewers, and the evaluation has been overseen by a Reviewing Editor and Patricia Wittkopp as the Senior Editor. The following individuals involved in review of your submission have agreed to reveal their identity: Dmitri A Petrov (Reviewer #1); Rama Ranganathan (Reviewer #2).

The reviewers have discussed the reviews with one another and the Reviewing Editor has drafted this decision to help you prepare a revised submission.

Summary:

The reviewers and I agreed that this is a very important paper that further identifies initial fitness as a key parameter that defines the rate of adaptation. Although the cause for this pattern remains mysterious, the current paper does help the field establish this pattern as one of few solid things we seem to have learned about the rate of adaptation.

We then agreed on the following list of revisions that we think are needed prior to publication:

Essential revisions:

Please include the starting fitness of each genotype as a covariate in the QTL models. Given that the authors had a strong expectation that the initial fitness would have a huge impact, and that in fact it did, we agreed that the key question is whether there is any heritability of fitness gain after correcting for the initial fitness.

The same strains evolved in different environments should be directly compared with each other, especially in cases when their initial fitnesses in the two environments was as different as possible within the spread of Figure 1. Keeping the genetic background constant and just changing the initial fitness should be very revealing and a powerful additional test of the core hypothesis. Given the competition experiments already performed directly comparing growth of at least some pairs of strains seems like it should be doable within a few months. We see this experiment as another way to validate the conclusions presented.

A concern was raised that the rest of the findings are somewhat underwhelming and feel underpowered by having too few crosses, too few segregants, too few mutations, lack of the ability to measure the fitness of each mutation, and certainly having too few environments. Despite these concerns, we felt the core findings were impactful enough to justify publication. We therefore leave it to the authors discretion to decide whether to keep or remove these analyses, but if kept, additional clarity that stipulates the limited conclusions that should be drawn from these data must be added. Adding additional data to support these points would also be welcome.

Please add the caveat that the central conclusions of extent and molecular pattern of heritability in adaption and pleiotropy will likely depend on the specific nature of the starting genotypes and the environmental conditions used.

We discussed the fact that testing some of the molecular insights about adaptability and pleiotropy through construction of strains that carry specific mutations or combinations of mutations in target genes in specific founder background would be greatly beneficial and would welcome any additional information you might have along these lines. We also agreed that if introducing specific nucleotide changes isn't feasible at this time, a comparison of alleles using a hemizygosity test or by swapping a larger region of DNA between RM and BY would still be very helpful. Experimental validation on at least one QTL (e.g., KRE33) is important to persuade readers that QTLs identified by these complicated statistical methods are genuine. All of that said, the majority opinion of the discussants was ultimately that this functional validation was not strictly needed for the current study to have a significant impact on the field, so we leave this to the author's discretion.

The concepts of microscopic and macroscopic epistasis are clearly stated but it is less obvious that the data support a binary classification of this property into these two categories. Does microscopic epistasis imply the local, low-order epistasis of mutations such that the effect does not propagate to globally influence fitness of other mutations? And is macroscopic epistasis just the case of a high-order epistasis such that mutations have more global influence? And then why could there not be a continuum of such influences between these extremes? Please clarify.

---

## [Author Response]

*Essential revisions:*

*Please include the starting fitness of each genotype as a covariate in the QTL models. Given that the authors had a strong expectation that the initial fitness would have a huge impact, and that in fact it did, we agreed that the key question is whether there is any heritability of fitness gain after correcting for the initial fitness.*

We entirely agree that a key question is about heritability of fitness gains after correcting for initial fitness. We therefore do include initial fitness as a covariate in the QTL models. This is described in the ‘Combined QTL and fitness model’ section of the Materials and methods, and is shown in Figure 3 (model fit parameters in the Figure 3—source data 1). We have modified the main text to clarify this issue further (subsection “Predicting adaptability from genotype and founder fitness”, paragraph 4). Also note that the fraction of heritable variation that is not accounted for by initial fitness is quoted in paragraph 3 of the same section.

We do note that it was not obvious to us a priori that fitness would be a good predictor of adaptability, given the large genetic distances between founders. Multiple scenarios seemed plausible. We therefore think that it is necessary to retain the discussion of the overall heritable variation in adaptability, as well as the fraction that can be explained by QTLs alone and fitness alone, as a point of comparison.

*The same strains evolved in different environments should be directly compared with each other, especially in cases when their initial fitnesses in the two environments was as different as possible within the spread of Figure 1. Keeping the genetic background constant and just changing the initial fitness should be very revealing and a powerful additional test of the core hypothesis. Given the competition experiments already performed directly comparing growth of at least some pairs of strains seems like it should be doable within a few months. We see this experiment as another way to validate the conclusions presented.*

This is a great suggestion. We did in fact evolve the same strains (founders) in replicate in each environment, which enables us to assess the fitness gains made by particular founders that had unusually disparate fitnesses in the two conditions. As the reviewers describe, the declining adaptability hypothesis predicts that founders with unusually high fitnesses at OT and low fitnesses at HT should adapt more at HT than at OT (and similarly for founders with low fitnesses at HT and high fitnesses at OT). We have conducted this analysis as suggested, and find the expected pattern. In the revised manuscript, we have added a new figure (new Figure 4) and a new section (‘Adaptability depends on fitness in the “home” environment’) to the main text describing these results.

*A concern was raised that the rest of the findings are somewhat underwhelming and feel underpowered by having too few crosses, too few segregants, too few mutations, lack of the ability to measure the fitness of each mutation, and certainly having too few environments. Despite these concerns, we felt the core findings were impactful enough to justify publication. We therefore leave it to the authors discretion to decide whether to keep or remove these analyses, but if kept, additional clarity that stipulates the limited conclusions that should be drawn from these data must be added. Adding additional data to support these points would also be welcome.*

*Please add the caveat that the central conclusions of extent and molecular pattern of heritability in adaption and pleiotropy will likely depend on the specific nature of the starting genotypes and the environmental conditions used.*

We agree with the reviewers that a thorough discussion of caveats and limitations of our study is important. This is particularly true since the nature of the phenotype we are studying (which requires long-term evolution experiments to measure) limits the number of crosses and segregants we can practically analyze. We have mentioned these limitations at appropriate points throughout the main text, and expanded the discussion of these issues in the Discussion section. We have also modified relevant section headers.

*We discussed the fact that testing some of the molecular insights about adaptability and pleiotropy through construction of strains that carry specific mutations or combinations of mutations in target genes in specific founder background would be greatly beneficial and would welcome any additional information you might have along these lines. We also agreed that if introducing specific nucleotide changes isn't feasible at this time, a comparison of alleles using a hemizygosity test or by swapping a larger region of DNA between RM and BY would still be very helpful. Experimental validation on at least one QTL (e.g., KRE33) is important to persuade readers that QTLs identified by these complicated statistical methods are genuine. All of that said, the majority opinion of the discussants was ultimately that this functional validation was not strictly needed for the current study to have a significant impact on the field, so we leave this to the author's discretion.*

We agree that more direct functional validation would be ideal, and in light of this we certainly agree that reconstruction experiments would be very valuable. We have added a brief discussion of this to the text (final paragraph of Discussion section). Unfortunately, however, we cannot independently validate the effect of KRE33 (or other QTLs) with simple allele swap experiments. The problem is that the trait of interest is adaptability above and beyond fitness, but KRE33 and all other significant adaptability QTLs also have a significant effect on fitness. This means that strains that differ only at this locus will start with different fitnesses, and it will not be possible to separate the effect of fitness and the focal allele. One could imagine addressing this by “balancing” the fitness of reconstructed strains by introducing additional fitness-affecting mutations. However, these balancing mutations might also have other uncontrolled effects on adaptability. Thus (in addition to being very involved) the results of these experiments may not be much cleaner or more direct than our existing results. We have therefore not pursued this avenue further here.

*The concepts of microscopic and macroscopic epistasis are clearly stated but it is less obvious that the data support a binary classification of this property into these two categories. Does microscopic epistasis imply the local, low-order epistasis of mutations such that the effect does not propagate to globally influence fitness of other mutations? And is macroscopic epistasis just the case of a high-order epistasis such that mutations have more global influence? And then why could there not be a continuum of such influences between these extremes? Please clarify.*

We agree that microscopic and macroscopic epistasis are neither nested nor mutually exclusive, and the reviewers are right that this is not a binary classification (there can certainly be a continuum of influences between the extremes). Our intention was instead to define these two concepts as the relevant aspects of epistasis that are important for answering different types of evolutionary questions. We have modified the Introduction accordingly to clarify these points (paragraph 4).